# Targeting therapy-resistant lung cancer stem cells via disruption of the AKT/TSPYL5/PTEN positive-feedback loop

In-Gyu Kim [1,2✉], Jei-Ha Lee[1], Seo-Yeon Kim[1], Chang-Kyu Heo[3], Rae-Kwon Kim[1,2] & Eun-Wie Cho [3✉]

Cancer stem cells (CSCs) are regarded as essential targets to overcome tumor progression and therapeutic resistance; however, practical targeting approaches are limited. Here, we identify testis-specific Y-like protein 5 (TSPYL5) as an upstream regulator of CSC-associated genes in non-small cell lung cancer cells, and suggest as a therapeutic target for CSC elimination. TSPYL5 elevation is driven by AKT-dependent TSPYL5 phosphorylation at threonine-120 and stabilization via inhibiting its ubiquitination. TSPYL5-pT120 also induces nuclear translocation and functions as a transcriptional activator of CSC-associated genes, *ALDH1* and *CD44*. Also, nuclear TSPYL5 suppresses the transcription of *PTEN*, a negative regulator of PI3K signaling. TSPYL5-pT120 maintains persistent CSC-like characteristics via transcriptional activation of CSC-associated genes and a positive feedback loop consisting of AKT/TSPYL5/PTEN signaling pathway. Accordingly, elimination of TSPYL5 by inhibiting TSPYL5-pT120 can block aberrant AKT/TSPYL5/PTEN cyclic signaling and TSPYL5-mediated cancer stemness regulation. Our study suggests TSPYL5 be an effective target for therapy-resistant cancer.

[1] Department of Radiation Biology, Environmental Radiation Research Group, Korea Atomic Energy Research Institute, Daejeon, South Korea. [2] Department of Radiation Science and Technology, Korea University of Science and Technology, Daejeon, South Korea. [3] Rare Disease Research Center, Korea Research Institute of Bioscience and Biotechnology, Daejeon, South Korea. ✉email: igkim@kaeri.re.kr; ewcho@kribb.re.kr

The development of radiotherapy and targeted drugs has significantly progressed cancer therapy. However, cancer metastasis and recurrence by inherent or acquired resistance mechanisms still occur at high rates. Despite ongoing efforts to improve the efficacy of cancer treatments, the challenge of overcoming resistance to radiotherapy and targeted cancer drugs remains a problem. Growing evidence suggests that CSCs play a pivotal role in inherent or acquired resistance to $\gamma$-radiation and targeted cancer drugs, such as epidermal growth factor receptor (EGFR)-tyrosine kinase inhibitors (TKIs)[1–3]. According to the CSC model, tumor initiation, growth, and maintenance are all driven by a small population of cells, known as tumor-initiating CSCs[4]. CSCs or dedifferentiated CSC-like cells are derived from nontumorigenic, differentiated cancer cells in the tumor microenvironment, and this process involves various factors, such as hypoxia and cytokine networks[5]. Similar to normal stem cells, CSCs exhibit self-renewal ability and can differentiate into various phenotypic progeny, which facilitates radio- or chemotherapeutic resistance and causes tumor relapse[6,7].

The presence of CSCs was first shown by surface markers, such as CD34+/CD38−, in a small subset of acute myeloid leukemia cells[8,9]. Subsequent studies showed that high levels of CD44, CD133, and CD326 (EpCAM) are also responsible for self-renewal in various solid tumors[10–13]. CSC-associated surface markers are involved in intracellular signaling cascades via the binding of specific ligands. For example, CD44 binds to hyaluronic acid or osteopontin to maintain a CSC phenotype and promote epithelial-to-mesenchymal transition (EMT), which is required for radiation or drug resistance as well as metastasis in cancer cells[14–17]. High ALDH1 activity is also an in vitro and in vivo hallmark of CSCs. Patients with tumor cells with high ALDH1 levels often exhibit therapeutic resistance and have a poor prognosis[18–21]. So far, intensive research has been conducted to identify and target CSCs using specific marker subsets[22]. Nevertheless, the targeted elimination of CSCs remains a challenge. To effectively eradicate CSCs, novel molecular targets, such as critical signaling components or CSC therapy-resistance markers, need to be further explored.

Testis-specific Y-like protein 5 (TSPYL5) is a potent oncogenic driver in some types of cancer[23–27]. In poor prognostic breast cancer, TSPYL5 promotes the degradation of the p53 tumor suppressor by directly inhibiting ubiquitin carboxyl-terminal hydrolase 7 (USP7), a p53 deubiquitinase[23]. The degradation of p53 associated with TSPYL5/USP7 complex formation also occurs in lung cancer cells that overexpress MUC16, which induces cisplatin and gemcitabine ressistance[24]. TSPYL5 is also crucial for the survival of ALT+ cancer cells. As a component of PML body in ALT+ cells, TSPYL5 prevents the polyubiquitination of POT1 via inhibition of USP7, of which functions as an activator of POT1 E3 ubiquitin ligase. Accordingly, TSPYL5 is suggested as an ALT+ cancer-specific therapeutic target[25]. In postmenopausal breast cancer, TSPLY5, as a nucleosome assembly protein, upregulates the transcription of several genes, such as CYP19A1[26]. Despite these critical functions of TSPYL5 in cancer, the studies evaluating TSPYL5 as an oncogenic driver are still limited. Also, its post-translational modifications have not been characterized.

In our previous study, we found that TSPYL5 is associated with $\gamma$-radiation resistance in non-small-cell lung cancer (NSCLC) cells through regulation of the PTEN/AKT pathway[27]. Also, it is well known that the PTEN/PI3K/AKT pathway is involved in the acquisition of resistance to EGFR-TKIs[28–30], and plays a pivotal role in CSC maintenance[31,32]. Based on these findings, we speculate that TSPYL5 may enhance CSC-like properties that promote $\gamma$-radiation and drug resistance in cancer cells via the PTEN/AKT signaling pathway. Here, we demonstrate that TSPYL5 mediates the AKT/TSPYL5/PTEN signaling loop and that AKT-mediated phosphorylation of TSPYL5 persistently maintains cancer stemness via transcriptional activation of the CSC-associated genes, ALDH1 and CD44. Also, we show that TSPYL5 can potentially be used as a CSC target for cancer therapy.

## Results

**TSPYL5 is upregulated in $\gamma$-radiation-exposed or gefitinib-treated NSCLC cells that show CSC-like properties.** The relevance of TSPYL5 expression to radiation resistance of CSC-like NSCLC cells has been examined with NSCLC cells (A549 human lung adenocarcinoma cell line; H460 human large-cell lung carcinoma cell line) treated with fractionated $\gamma$-radiation[33] [2 (or 1) Gy × 3 with a 3-day interval, a total of 6 (or 3) Gy]. Exposure of NSCLC cells to $\gamma$-radiation increased the cellular levels of ALDH1 activity and CD44 expression (Fig. 1a and Supplementary Fig. 1a), which are representative biomarkers that impart chemo- or radio-resistant CSC properties to cancer cells[34–36]. These radiation-exposed cells exhibited a high sphere-formation ability, which represents the self-renewal potential (Fig. 1b). Radiation also activated the EMT program as shown by invasion and migration assay (Fig. 1c). CSC-like properties of these cells were further demonstrated by the increased expression of CSC biomarkers, including ALDH1 (ALDH1A1 and ALDH1A3 isoforms), CD44, SOX-2, and OCT-4 (Fig. 1d and Supplementary Fig. 1b). EMT markers, such as N-cadherin, Snail, and Vimentin, were also increased, thus providing a distinct advantage for metastatic dissemination (Fig. 1d). Cells that survived even after the higher dose of fractionated $\gamma$-irradiation (2 Gy × 10 with a 3-day interval, a total dose of 20 Gy) also showed the increase of CSC and EMT properties as demonstrated by specific markers (Supplementary Fig. 1c). Interestingly, in all of these radiation-exposed CSC-like cells with low or high dose, TSPYL5 expression increased in proportion to the radiation dose. Gefitinib-resistant H460 cells (GR)[37–39], which were selected by a stepwise escalation of EGFR TKI gefitinib treatment, also showed CSC-like properties: ALDH1 activity and CD44 expression were increased (Fig. 1e), and the sphere formation and metastatic capacities were enhanced (Fig. 1f, g). TSPYL5 also upregulated in these cells along with the ALDH1 and CD44 elevation (Fig. 1h).

To further analyze the relevance of TSPYL5 in lung CSCs, CSC-like subpopulations were sorted from A549 NSCLC cells via Aldefluor staining (Fig. 2a). The subpopulation of CSC-like cells with high ALDH1 expression (AL$^{high}$) showed typical properties of self-renewal potential and invasion/migration capacity (Fig. 2b). Cancer stemness markers (ALDH1, CD44, OCT-4, and SOX-2) highly increased when comparing to the cell subpopulation with low ALDH1 expression (AL$^{low}$) (Fig. 2c). The ALDH1$^{high}$ subpopulation sorted from H460 cells also showed typical CSC-like characteristics and the expression of cellular markers (Fig. 2d–f). Similar to the radiation-exposed and gefitinib-resistant cells, both ALDH1$^{high}$ CSC-like subpopulations (A549 and H460) showed higher levels of TSPYL5 compared with the corresponding ALDH1$^{low}$ cells (Fig. 2c, f). CD44$^{high}$ cells sorted from A549 NSCLC cells also showed stem-like and metastatic properties with high levels of ALDH1 and TSPYL5 compared with CD44$^{low}$ cells (Fig. 2g–i).

Collectively, radiation-exposed and gefitinib-resistant NSCLC cells showed cellular characteristics similar to CSC-like ALDH1$^{high}$ or CD44$^{high}$ cells, which all expressed high levels of TSPYL5. These results suggest an association among ALDH1, CD44, and TSPYL5 expression in response to radiation, gefitinib, or cancer stemness.

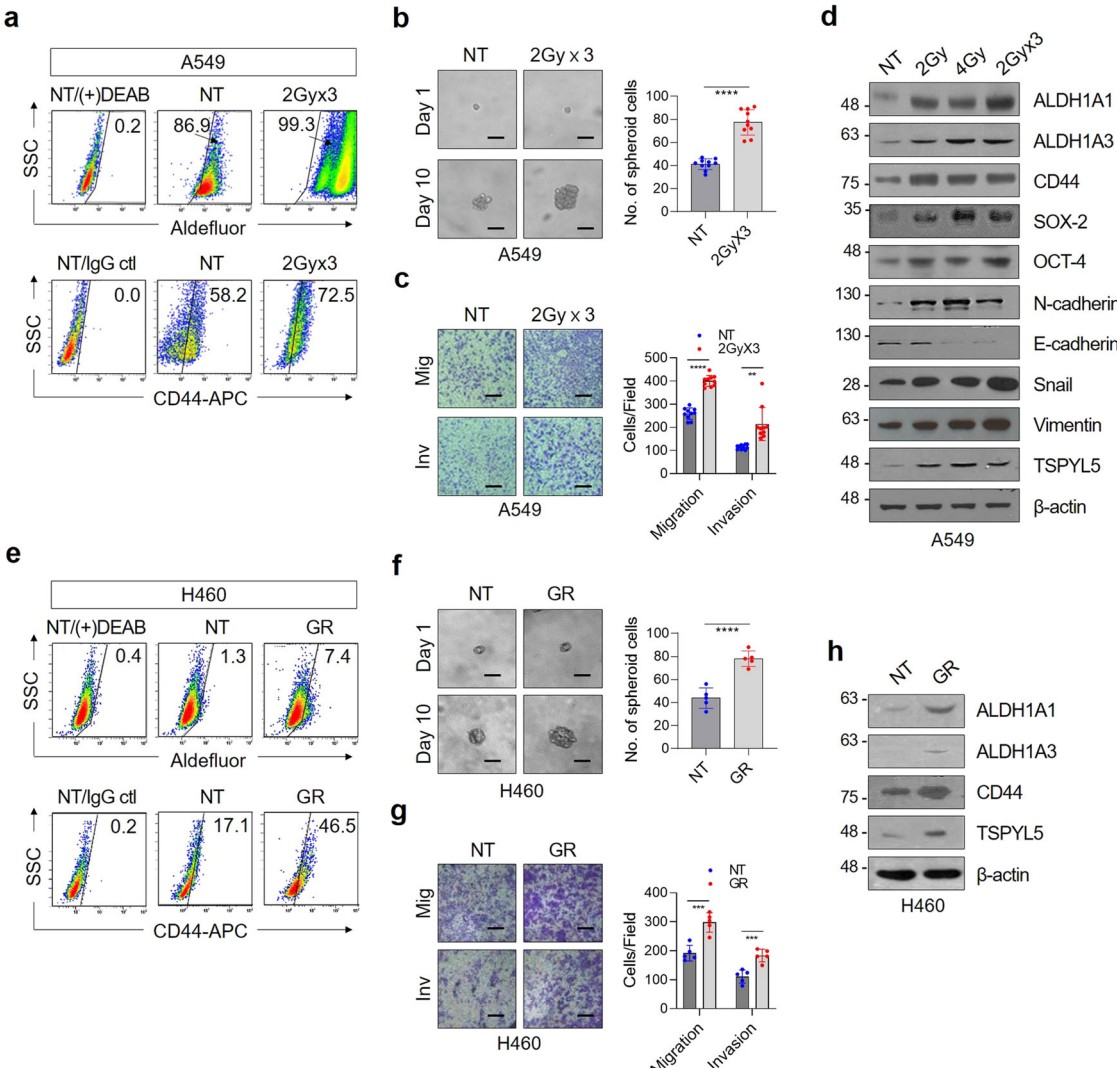

**Fig. 1 Fractionated γ-irradiation or gefitinib treatment increases TSPYL5 expression in NSCLC cells. a** Flow cytometric analysis of ALDH1 activity (Aldefluor staining) and CD44 expression (CD44-APC) in A549 cells exposed to fractionated γ-irradiation. Cells were irradiated with a total dose of 2, 4, or 6 Gy (2 Gy × 3) using $^{60}$Co γ-ray at a dose rate of 0.2 Gy/min. N,N-diethylaminobenzaldehyde (DEAB)-treated cells or isotype control IgG-stained cells were used as negative controls for gating. Numbers in the upper-right corner represent the percent of positively stained cells. $n = 10$ independent experiments for each group. **b** Sphere-formation assay of fractionated γ-irradiation-exposed A549 cells. $n = 5$. **c** Migration/invasion assay of fractionated γ-irradiation-exposed A549 cells. $n = 5$. Scale bar: 20 μm. **d** Western blot analysis of CSC (ALDH1A1 and A3, CD44, OCT-4, and SOX-2), EMT (N-cadherin, E-cadherin, Snail, and Vimentin) markers, and TSPYL5 in γ-irradiation-exposed A549 cells. CD44 detected with pan-CD44 antibody [CD44 (8E2), CST] corresponds to CD44 standard (CD44s) with molecular weight of about 85 kDa. **e** Flow cytometric analysis of CSC markers (ALDH1 activity and CD44 expression) in gefitinib-resistant (GR) H460 cells. GR-H460 cells were selected by stepwise escalation of gefitinib from 1 to 10 μmol/L over 4 months. **f** Sphere-formation assay of GR-H460 cells. $n = 5$. **g** Migration/invasion assay of GR-H460 cells. $n = 5$. **h** Western blot analysis of CSC markers (ALDH1A1, ALDH1A3, and CD44) and TSPYL5 in GR-H460 cells. Data represent mean ± s.d. using two-tailed t-test. $^{**}p < 0.01$, $^{***}p < 0.001$, $^{****}p < 0.0001$.

**TSPYL5 enhances CSC-like properties in NSCLC cells by promoting ALDH1 and CD44 and inhibiting PTEN expression.** To evaluate whether TSPYL5 substantially enhances CSC-like properties in radiation-exposed, gefitinib-treated, or in ALDH1$^{high}$ cells, we examined alterations in self-renewal and EMT potential in response to TSPYL5 overexpression or knockdown. In A549 cells, which have high endogenous TSPYL5 levels (Supplementary Fig. 1d), TSPYL5 knockdown by siRNA markedly reduced sphere formation and decreased cellular invasiveness and motility (Fig. 3a). The spindle-shaped cell morphology of A549 cells also had returned to a cobblestone-like shape, which is characteristic of differentiated cells. With these cellular changes in self-renewal and metastatic capacities, the levels of CSC-associated factors (OCT-4 and SOX-2) and EMT

markers (N-cadherin and Vimentin) were decreased (Fig. 3b). In contrast, TSPYL5 overexpression in H460 cells, which have low endogenous TSPYL5 levels (Supplementary Fig. 1d), enhanced the sphere formation and metastatic capacities and induced morphological changes from a cobblestone-like shape to a spindle shape, which is a typical CSC or EMT characteristic (Fig. 3c). The elevation of EMT and CSC-associated factor levels was also confirmed by Western blotting (Fig. 3d).

We further explored the association of TSPYL5 with the expression of ALDH1 and CD44. Knockdown of TSYPL5 expression in A549 cells resulted in downregulation of ALDH1 and CD44 (Fig. 3e), whereas TSPYL5 overexpression increased the expression of ALDH1 and CD44 in H460 cells (Fig. 3f). TSPYL5 altered ALDH1 and CD44 at the transcriptional levels,

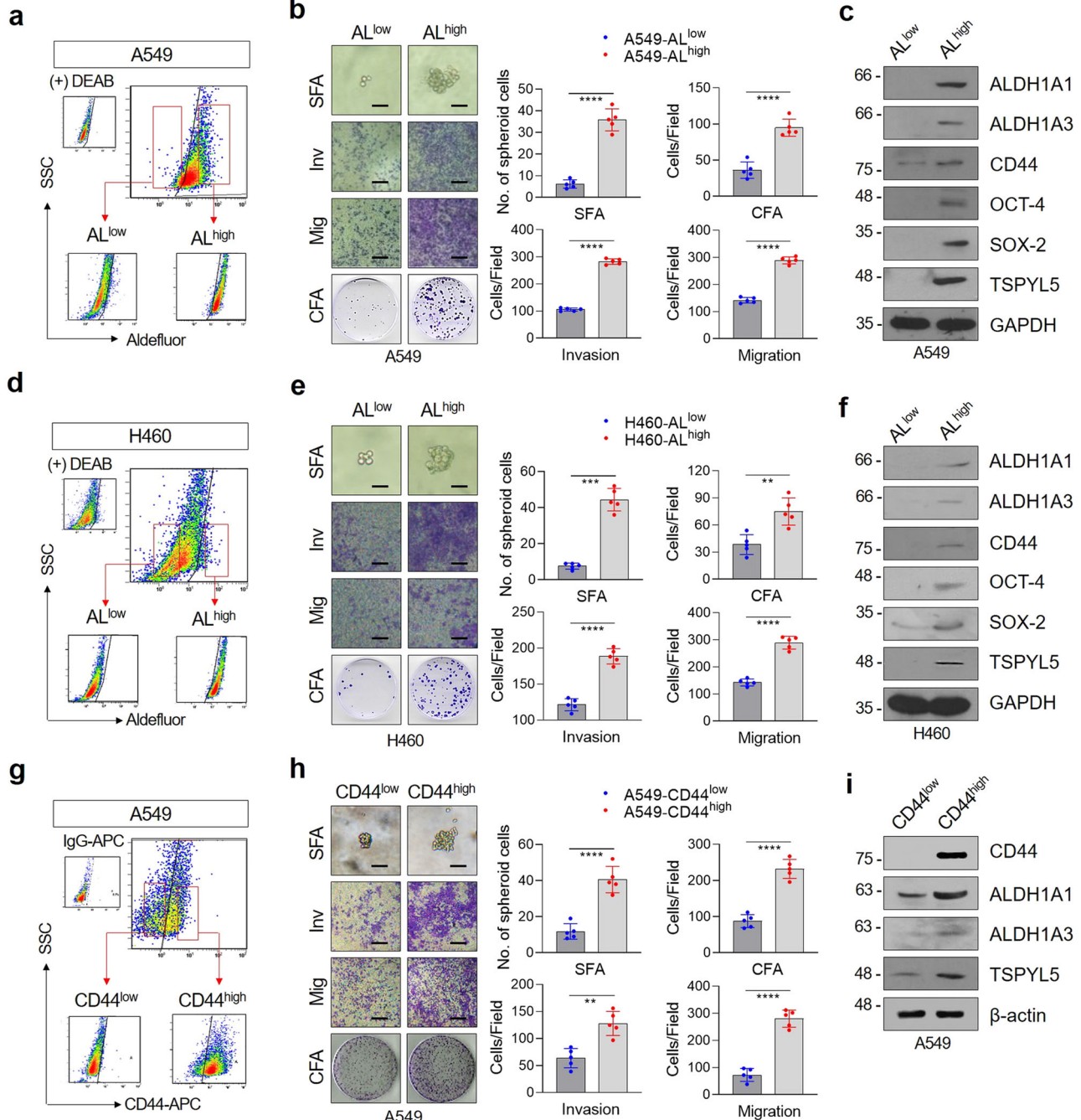

**Fig. 2 The expressions of TSPYL5 and CD44 are elevated in ALDH1$^{high}$ CSC-like cells isolated from NSCLC cells. a, d** Flow cytometric analysis of CSC-like cells using Aldefluor staining. A549 or H460 cells were stained with Aldefluor substrate and sorted into ALDH1$^{low}$ (AL$^{low}$) and ALDH1$^{high}$ (AL$^{high}$) cells. Sorted cells were re-examined by Aldefluor assay to confirm their ALDH1 activity. **b, e** Sphere formation, migration/invasion assay, and colony formation assay of AL$^{low}$ and AL$^{high}$ cells sorted from A549 or H460 cells. Scale bar: 20 μm. **c, f** Western blot analysis of CSC markers (ALDH1A1, ALDH1A3, CD44, OCT-4, and SOX-2) and TSPYL5 in AL$^{low}$ and AL$^{high}$ populations sorted from A549 or H460 cells. **g** Flow cytometric analysis of CSC-like cells using anti-CD44 immunostaining. A549 cells were stained with APC-conjugated anti-human CD44 antibody and sorted into CD44$^{low}$ and CD44$^{high}$ cells. Sorted cells were re-examined by CD44 staining to confirm their CD44 expression. **h** Sphere formation, migration/invasion assay, and colony-formation assay of CD44$^{low}$ and CD44$^{high}$ cells sorted from A549 cells. **i** Western blot analysis of CSC markers (ALDH1A1, ALDH1A3, and CD44) and TSPYL5 in CD44$^{high}$ and CD44$^{low}$ cells sorted from A549 cells. Data represent mean ± s.d. using two-tailed $t$-test. $n = 5$ independent experiments for each group. $^{*}p < 0.05$, $^{**}p < 0.01$, $^{***}p < 0.001$, $^{****}p < 0.0001$.

which subsequently influenced the corresponding protein expression levels (Fig. 3e, f). ALDH1 also can modulate the EMT-associated, CSC-like properties of the NSCLC cells: upregulation or knockdown of ALDH1A1 or ALDH1A3 influenced sphere formation and invasion/migration capacities as well as altered the

expression of the corresponding CSC and EMT markers (Supplementary Fig. 2a–d). However, differently from TSPYL5, the upregulation or knockdown of ALDH1A1 or ALDH1A3 did not alter the CD44 levels. TSPYL5 level was not changed also (Supplementary Fig. 2b, d), despite the changes in CSC-like

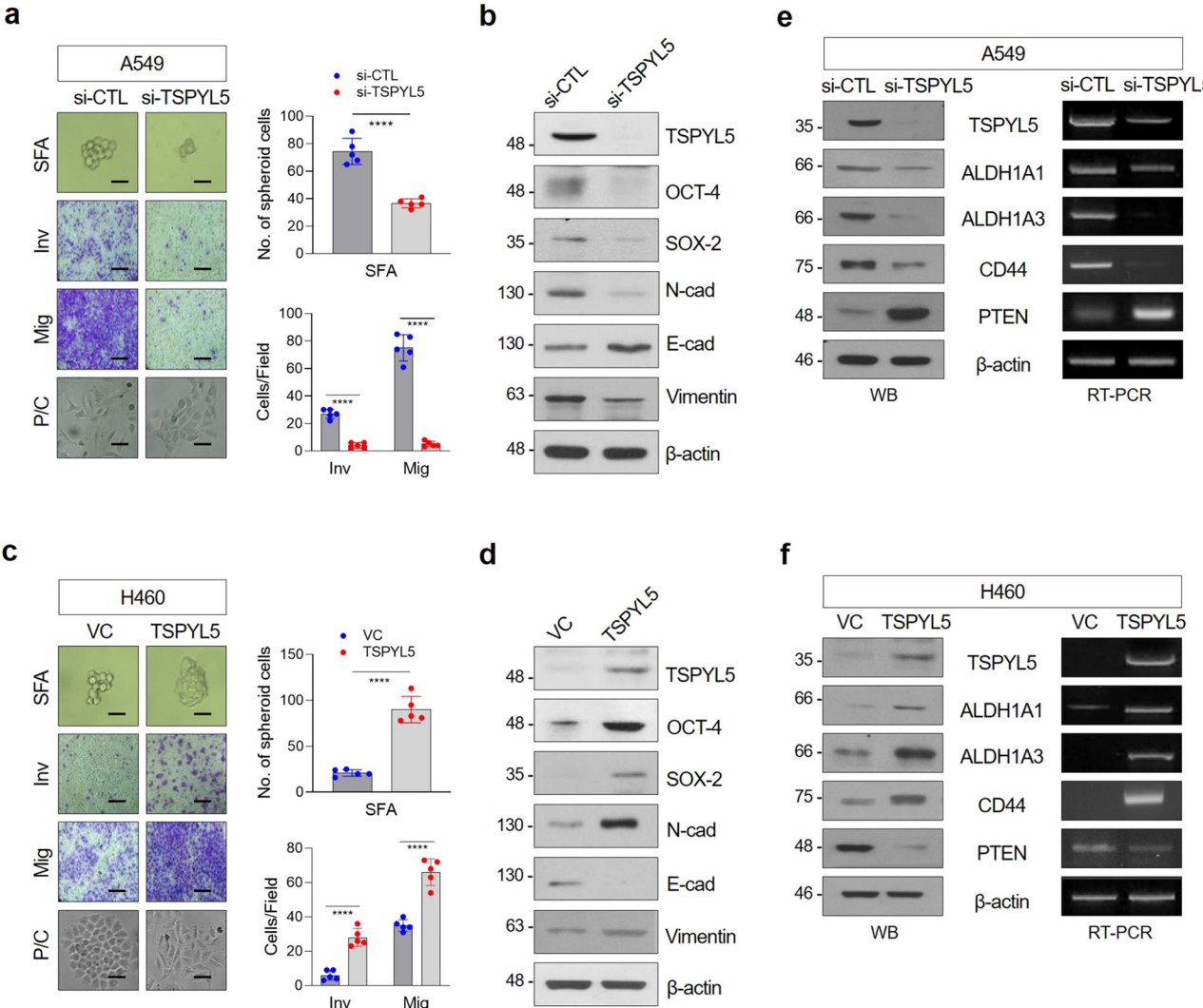

**Fig. 3 TSPYL5 promotes the stemness and EMT of cancer cells through upregulation of ALDH1 and CD44 levels and downregulation of PTEN level. a, c** Sphere formation, invasion/migration assay, and morphological observation indicated that the changes in stemness and EMT properties of cancer cells were affected by forced TSPYL5 suppression (si-TSPYL5) or overexpression (pcDNA3.1-TSPYL5) in A549 or H460 NSCLC cells. $n = 5$ independent experiments for each assay. Data represent mean ± s.d. using two-tailed $t$-test. $^{****}p < 0.0001$. Scale bar: 20 μm. **b, d** Western blot analysis of CSC (OCT-4, SOX-2) and EMT (N-cadherin, E-cadherin, and Vimentin) biomarkers in cancer cells affected by forced suppression or overexpression of TSPYL5. **e, f** Western blot and RT-PCR analysis of ALDH1 (ALDH1A1, ALDH1A3 isoforms), CD44 (CD44s corresponding to transcript variant 4), and PTEN protein and transcript levels following forced TSPYL5 suppression or overexpression.

properties (Supplementary Fig. 2a, c). These results implicate that all of them, ALDH1, CD44, as well as TSPYL5, can regulate cancer stemness; however, ALDH1 and CD44 are downstream regulatory targets of TSPYL5.

Also, we found that TSPYL5 negatively regulates PTEN expression (Fig. 3e, f), which antagonizes PI3K function by catalyzing PIP3 dephosphorylation, leading to AKT inactivation[40,41]. PTEN transcript and protein levels were down-regulated by TSLYL5 overexpression (Fig. 3e, f), as partially shown in our previous study[27].

**Loss of PTEN stabilizes TSPYL5 via AKT-dependent phosphorylation, which reinforces CSC properties and therapeutic resistance in NSCLC cells.** PTEN/PI3K/AKT pathway is involved in the acquisition of resistance to targeted drugs[28–30], and is crucial for CSC maintenance[31,32]. Decreased PTEN expression and consequent AKT activation was consistently observed in radiation-exposed, gefitinib-resistant, and CSC-like ALDH1[high]

NSCLC cells (Fig. 4a). Therefore, we evaluated the CSC-like properties and related gene expression levels in response to PTEN overexpression or AKT knockdown to confirm whether AKT activation is essential for these CSC-like properties in NSCLC cells. Sphere-formation assay showed that AKT knockdown inhibited the self-renewal potential of the cells (Fig. 4b, upper panels). Furthermore, AKT knockdown resulted in increased PTEN expression, whereas ALDH1 and CD44 expression was decreased (Fig. 4b, lower panels). In response to PTEN over-expression, AKT was in an inactive dephosphorylated state. Also, ALDH1 and CD44 expression was suppressed, which was also confirmed by the decrease of self-renewal potential (Fig. 4c). In these AKT-inactivated cells, TSPYL5 protein levels also decreased (Fig. 4b, c). However, TSPYL5 transcript levels were not altered in response to AKT knockdown or PTEN overexpression. In contrast, AKT overexpression or PTEN knockdown enhanced the sphere-formation ability of H460 cells (Fig. 4d, e). In these cells, AKT was activated by phosphorylation and the expression levels of ALDH1 and CD44 were increased. Also, TSPYL5 protein levels

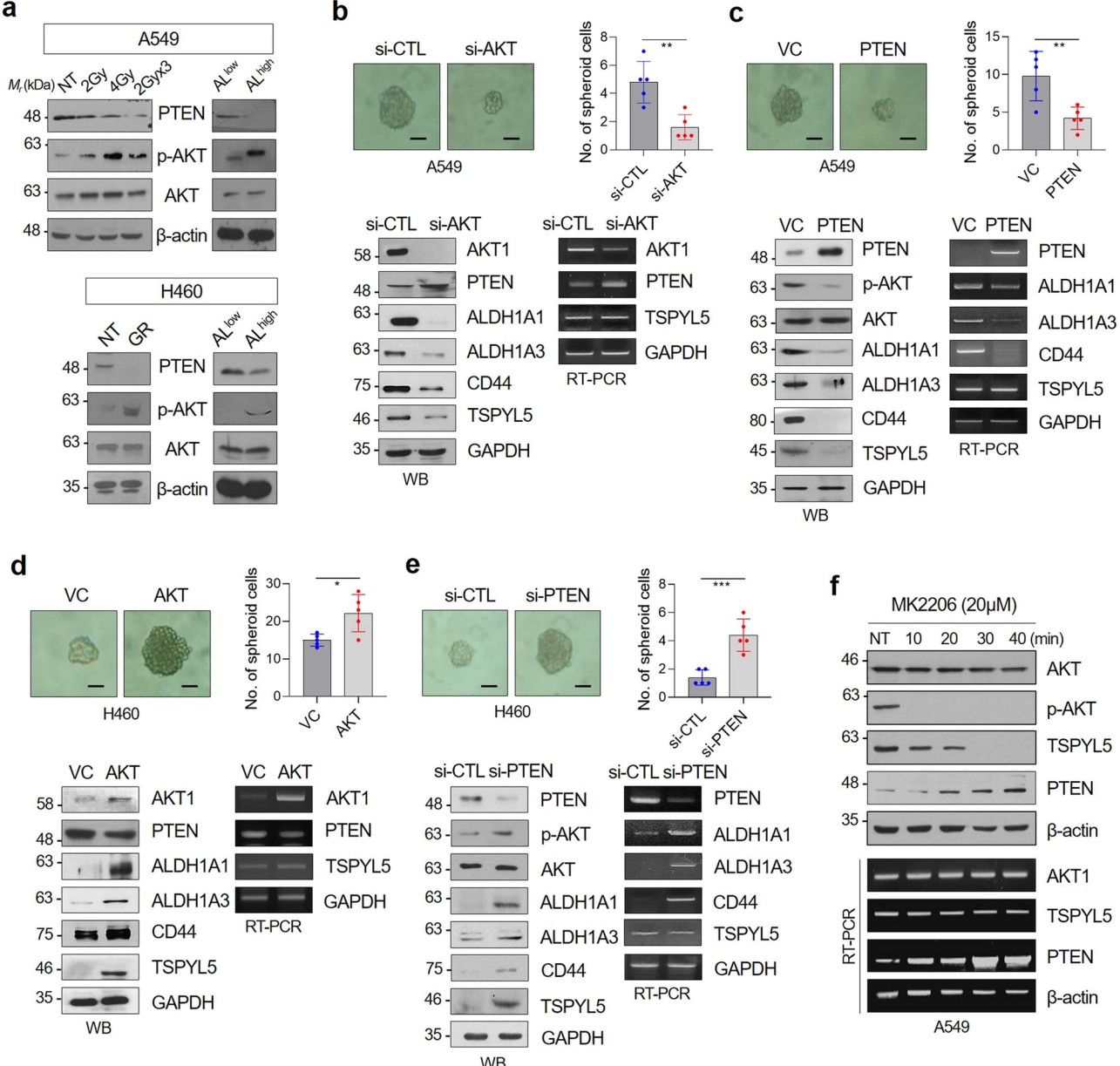

**Fig. 4 PTEN loss is associated with AKT activation as well as TSPYL5 expression in CSC-like lung cancer cells. a** PTEN loss and AKT activation were confirmed by Western blot analysis in γ-irradiation-exposed A549 cells (total dose of 2, 4, or 6 Gy [2 Gy × 3], using $^{60}$Co γ-ray at a dose rate of 0.2 Gy/min), gefitinib-resistant (GR) H460 cells, or sorted AL$^{high}$ NSCLC cells. NT: not treated. **b, c** CSC-associated sphere-forming ability and the expression of specific biomarkers (ALDH1A1/A3, CD44) and TSPYL5 in AKT-knockdown (si-AKT) or PTEN-overexpressing A549 cells. Scale bar: 20 μm. **d, e** CSC-associated sphere-forming ability and the expression of specific biomarkers (ALDH1A1/A3, CD44) and TSPYL5 in AKT-overexpressing or PTEN-knockdown (si-PTEN) H460 cells. n = 5 independent experiments for each group. Data represent mean ± s.d. using two-tailed t-test. *p < 0.05, **p < 0.01, ***p < 0.001. **f** Time-dependent effect of the AKT inhibitor MK2206 on the expression levels of TSPYL5 and PTEN. The transcript as well as protein levels of TSPYL5 and PTEN were determined by RT-PCR and Western blotting analysis.

were upregulated; however, its transcript levels were not altered again. Reduced protein levels of TSPYL5, but not its transcript level, were also confirmed in ALDH1$^{low}$ A549 cells compared with the CSC-like ALDH1$^{high}$ cells (Fig. 2c, Supplementary Fig. 3). These results imply that in CSC-like NSCLC cells, AKT activation by the loss of PTEN enhances TSPYL5 expression at the protein level, and it depends on AKT-dependent phosphorylation, whereas TSPYL5 regulates PTEN expression at the transcription level.

The regulation of TSPYL5 expression by AKT-dependent phosphorylation was further examined using an AKT inhibitor assay, in which AKT phosphorylation was completely suppressed

by treatment of MK2206. As expected, MK2206 treatment resulted in decreased TSPYL5 protein levels with unaltered TSPYL5 transcript levels (Fig. 4f). It showed that AKT-dependent phosphorylation stabilizes TSPYL5 protein, possibly by inhibiting its degradation. Interestingly, elevated PTEN expression was detected soon after the TSPYL5 protein levels were decreased in response to AKT inhibition. The self-renewal capacity of the A549 cells was also reduced by AKT inhibition (Supplementary Fig. 4), which may be associated with the decrease of TSPYL5.

Overall, aberrantly activated AKT in therapy-resistant or CSC-like cells has been shown to maintain the expression of TSPYL5 protein through AKT-dependent phosphorylation. From the

aspect of TSPYL5, it maintains AKT activation and itself via an AKT/TSPYL5/PTEN regulatory loop in a positive-feedback manner. Without AKT-dependent stabilization, TSPYL5 quickly disappeared, despite sufficient expression of its transcript. The decrease of TSPYL5 by inhibition of AKT consequently induced PTEN expression and again suppressed AKT activation. TSPYL5 acts as a CSC-associated factor by upregulating ALDH1 and CD44 expression, as well as by the AKT/TSPYL5/PTEN cyclic signaling loop.

**AKT-mediated phosphorylation of TSPYL5 at threonine-120 is essential for its stabilization and nuclear translocation.** Aberrant AKT activation caused by oncogenic mutations or PTEN loss is associated with poor prognosis in cancer[42,43]. Also, the activated AKT contributes to various cellular processes either directly or indirectly via post-translational modifications of substrate proteins, including phosphorylation, ubiquitination, or SUMOylation[44–48], which consequently determine the fate of the target proteins. The intracellular localization of TSPYL5 in response to AKT inhibition was examined by fluorescence microscopy in NSCLC cells to investigate the function of phosphorylated TSPYL5 as a CSC-associated factor. In A549 cells, TSPYL5 was highly expressed and localized in the nucleus as well as in the cytoplasm (Fig. 5a, upper panels). However, treatment with either MK2206 (AKT inhibitor) or LY294002 (PI3K inhibitor) significantly diminished the nuclear localization of TSPYL5 (Fig. 5a, low panels). Of course, AKT and PI3K inhibition decreased the total TSPYL5 levels, which were accompanied by CD44 and ALDH1 downregulation (Fig. 5b). Consistently, the nuclear accumulation of TSPYL5 was diminished in response to AKT or PI3K inhibition in H460 cells with TSPYL5 overexpression (Fig. 5c), thus resulting in its predominant cytoplasmic localization. TSPYL5 has been reported as a nucleosome protein[25]; however, the mechanism of its nuclear localization has not yet been elucidated. Our results suggest that AKT-mediated phosphorylation of TSPYL5 promotes not only its stabilization but also nuclear translocation. In addition, this leads us to speculate that nuclear TSPYL5 functions as a transcriptional regulator of specific genes that are important for characterizing EMT-associated, CSC-like properties, such as ALDH1 and CD44.

To further confirm the phosphorylation-dependent function of TSPYL5, the amino acid residue that is phosphorylated by AKT was examined. Until now, there have been no studies on the role of TSPYL5 post-translational modifications. However, proteomic studies have revealed several post-translational modification sites of TSPYL5, namely, four phosphorylation sites (T120[49], S180, S201, and S227[49,50]). When presuming phosphorylation sites using phosphorylation-site prediction software NetPhos 2.0 (http://www.cbs.dtu.dk/services/NetPhos), four threonine residues (T120, T177, T326, and T409) were expected to be potential phosphorylation sites of TSPYL5 with a 50–90% probability (Supplementary Table 1). Also, we compared the AKT substrate phosphorylation motif[51] **R**X**R**XXS/**T** with the whole TSPYL5 amino acid sequence, and found two potential protein sequence segments similar to the AKT substrate motif around S11 (RS**R**G**R**KSS[11]RAKNRGK) and T120 (SE**R**LAAD**T**[120]VFVG-TAG). Of these backgrounds, we considered the T120 residue of TSPYL5 as a potential AKT phosphorylation site and performed the mutant studies.

When non-mutagenic, wild-type TSPYL5 (WT-TSPYL5) was forcibly expressed in H460 cells, both nuclear and cytoplasmic TSPYL5 accumulation was observed, and the cellular levels of CD44 and ALDH1 were significantly upregulated (Fig. 5d, e). In contrast, the TSPYL5 alanine mutant (T120A-TSPYL5), of which potential phosphorylation site T120 is substituted with alanine,

showed cytoplasmic accumulation only, and CD44 and ALDH1 levels were not increased. Of course, total T120A-TSPYL5 expression was lower than that of WT-TSPYL5 (Fig. 5e, lower panel). The TSPYL5 aspartic acid mutant (T120D-TSPYL5), which is a mimic of phospho-T120-TSPYL5, acted similar to wild-type TSPYL5: T120D-TSPYL5 showed both nuclear and cytoplasmic localization, and CD44 and ALDH1 were elevated with its expression. The function and intracellular distributions of other threonine residue TSPYL5 mutants (T177A, T326A, and T409A) were not different from those of WT-TSPYL5 (Supplementary Fig. 5a, b and Fig. 5d, e). T120 mutation of TSPYL5 also showed suitable self-renewal and EMT potential in H460 cells (Supplementary Fig. 6). Altogether, these results indicate that TSPYL5 phosphorylation at T120 is essential for TSPYL5 stabilization and nuclear translocation as well as the subsequent expression of CD44 and ALDH1 in CSC-like NSCLC cells.

To further confirm the phosphorylation of TSPYL5 at T120, TSPYL5 was detected using anti-pT120-TSPYL5 antibody that was generated with a phospho-peptide containing pT120 of TSPYL5 (see "Methods"). Anti-TSPYL5 antibody detected WT-TSPYL5, T120A-TSPYL5, or T120D-TSPYL in the cell lysates of transfected H460 cells (Fig. 5f). However, the anti-pT120-TSPYL5 antibody detected WT-TSPYL5 and T120D-TSPYL5, but not T120A-TSPYL5. Additionally, after immunoprecipitation with antibodies against phosphorylated threonine or serine (p-T/S), only WT- and T120D-TSPYL5, but not T120A-TSPYL5, were detected with anti-pT120-TSPYL5 antibody (Fig. 5f).

Immunoprecipitation assay with anti-AKT antibody showed that T120 of TSPYL5 is a target of AKT. Endogenous TSPYL5 and AKT were co-immunoprecipitated with specific antibodies of each other in A549 cells (Fig. 5g). Exogenously overexpressed TSPYL5 was also co-immunoprecipitated with AKT in H460 cells (Fig. 5h). However, mutant T120A-TSPYL5 did not precipitate AKT.

**AKT-mediated phosphorylation of TSPYL5 at threonine-120 blocks its ubiquitination and induces its SUMOylation.** TSPYL5 stability was analyzed after treatment with MK2206 and proteasome inhibitor MG132 to examine whether T120 phosphorylation stabilizes TSPYL5 by interfering with its degradation process (Fig. 6a). Upon its overexpression in H460 cells, elevated WT-TSPYL5 levels were detected along with activated AKT (p-AKT). WT-TSPYL5 was reduced by MK2206 treatment; however, MG132 blocked its reduction. T120A-TSPYL5, an unstable TSPYL5 mutant, was not observed even when forcibly expressed, whereas it was detected after MG132 treatment. T120D-TSPYL5, a mimic of phospho-T120-TSPYL5, showed more elevated expression than WT-TSPYL5, and, even when MK2206 was treated, the remaining TSPYL5 was detected. These results suggest that AKT-dependent phosphorylation of TSPYL5 at T120 stabilizes it through inhibition of proteasomal degradation, which may be mediated by inhibition of ubiquitination. It was further confirmed by an in vivo ubiquitination assay (Fig. 6b). MK2206 treatment enhanced the ubiquitination of WT-TSPYL5 and its consequent degradation. T120D-TSPYL5 and T409A-TSPYL5 also showed similar ubiquitination and degradation patterns to those of WT-TSPYL5. However, T120A-TSPYL5 mutant, which cannot be a substrate for AKT-dependent phosphorylation, displayed elevated ubiquitination and degradation, even without AKT inhibition.

SUMOylation often functions as a regulatory mechanism for the transport of cellular proteins between the cytoplasm and nucleus[52]. We examined whether the SUMOylation of TSPYL5 is associated with its nuclear translocation and it depends on T120

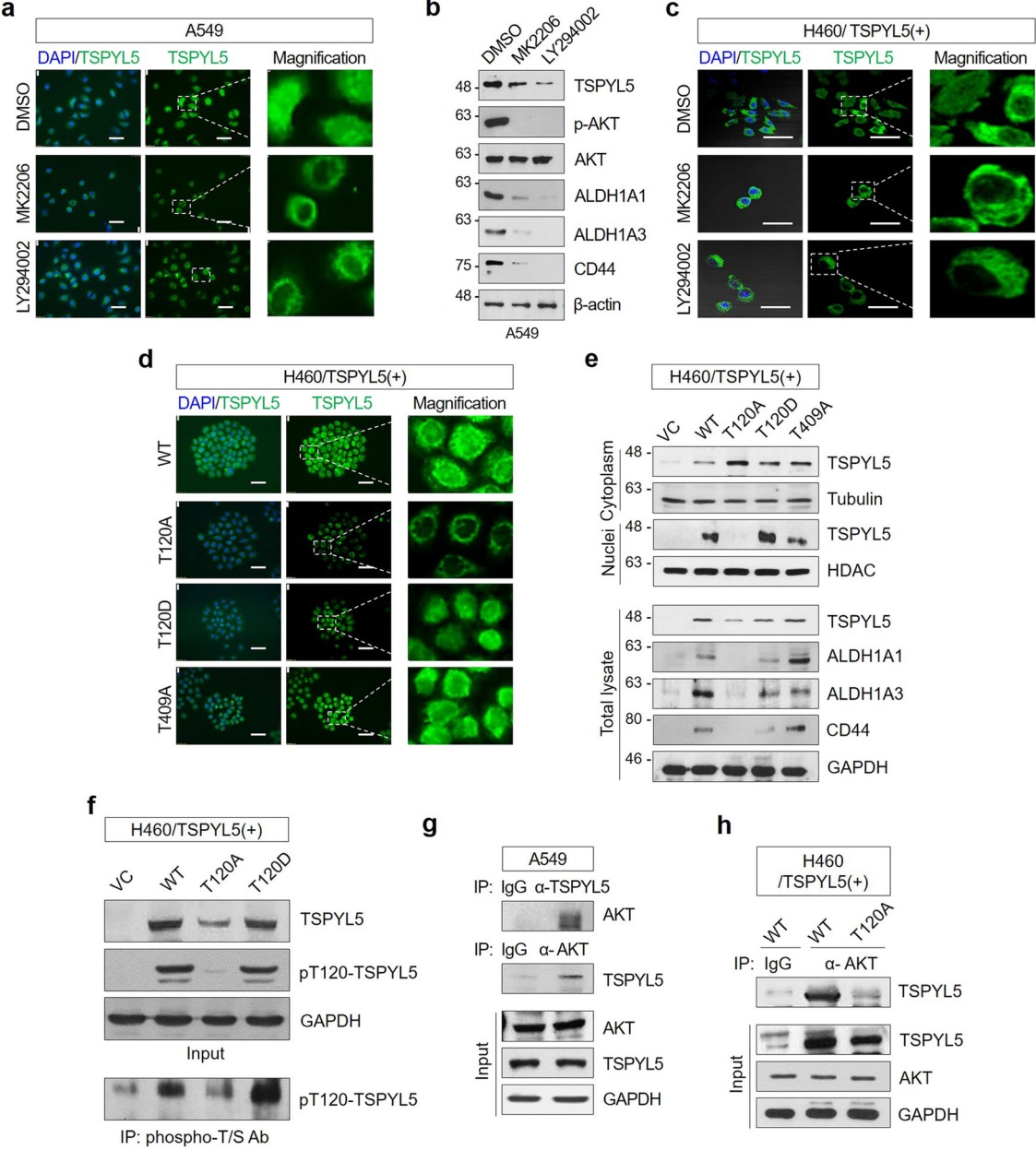

**Fig. 5 AKT-mediated phosphorylation of TSPYL5 results in stabilization and nuclear translocation of TPYL5 and enhancement of CSC-like characteristics.** **a** Reduction of nuclear TSPYL5 in A549 cells by treatment of the AKT inhibitor MK2206 (20 μM) or the PI3K inhibitor LY294002 (10 μM) for 4 h. Inserts of dashed-line boxes were magnified in the right panel. Scale bar: 20 μm. **b** Expression of TSPYL5 and CSC markers (ALDH1A1/A3, CD44) in A549 cells treated with AKT or PI3K inhibitors. **c** Reduction of nuclear TSPYL5 in TSPYL5-overexpressing H460 cells by treatment of the AKT inhibitor MK2206 or the PI3K inhibitor LY294002. **d** Intracellular localization of TSPYL5 mutants in H460 cells (WT: non-mutagenic TSPYL5 with T120, T120A: alanine mutant of T120 of TSPYL5, T120D: aspartic acid mutant of T120, and T409A: alanine mutant of T409). **e** Reduction and cytoplasmic retention of the TSPYL5-T120A mutant and suppression of ALDH1 and CD44 in H460 cells. The total cell lysates of H460 cells transfected with WT or site-directed mutagenic TSPYL5-expression vectors were fractionated into nuclei and cytoplasm, and TSPYL5 expressions were analyzed by Western blotting. Tubulin and HDAC were used as fraction-specific markers. CSC markers (ALDH1A1/A3, CD44) were analyzed in TSPYL5-mutant-expressing H460 cells. VC: pcDNA3.1 vector-transfected control. **f** Phosphorylation of T120 in TSPYL5. Cell lysates from H460 cells transfected with non mutagenic (WT: T120) or site-directed mutagenic TSPYL5-expression (T120A or T120D) vectors or their immunoprecipitates with anti-phospho-threonine/serine (T/S) antibody were analyzed with anti-pT120-TSPYL5 antibody (see "Methods"). **g, h** Coimmunoprecipitation assay showing a direct interaction between AKT and TSPYL5 in A549 cell lysates and H460 cells transfected with WT-TSPYL5 or T120A-TSPYL5 expression vector. Immunoprecipitations were performed with anti-TSPYL5 antibody or anti-AKT antibody. Immunoprecipitates and total cell lysates were analyzed by Western blotting analysis.

phosphorylation. When A549 cells were treated with ginkgolic acid, a SUMOylation inhibitor that blocks the formation of the E1-SUMO intermediate[53], nuclear TSPYL5 significantly disappeared (Fig. 6c). Moreover, an in vivo SUMOylation assay revealed that WT-TSPYL5 was significantly conjugated with

SUMO1, whereas the T120A-TSPYL5 mutant was not (Fig. 6d). These results suggest that SUMOylation of TSPYL5 is essential for its nuclear entry and that phosphorylation of TSPYL5 at T120 is required for its SUMOylation. TSPYL5 mutants with an alanine substitution at T177, T326, or T409, which were all translocated

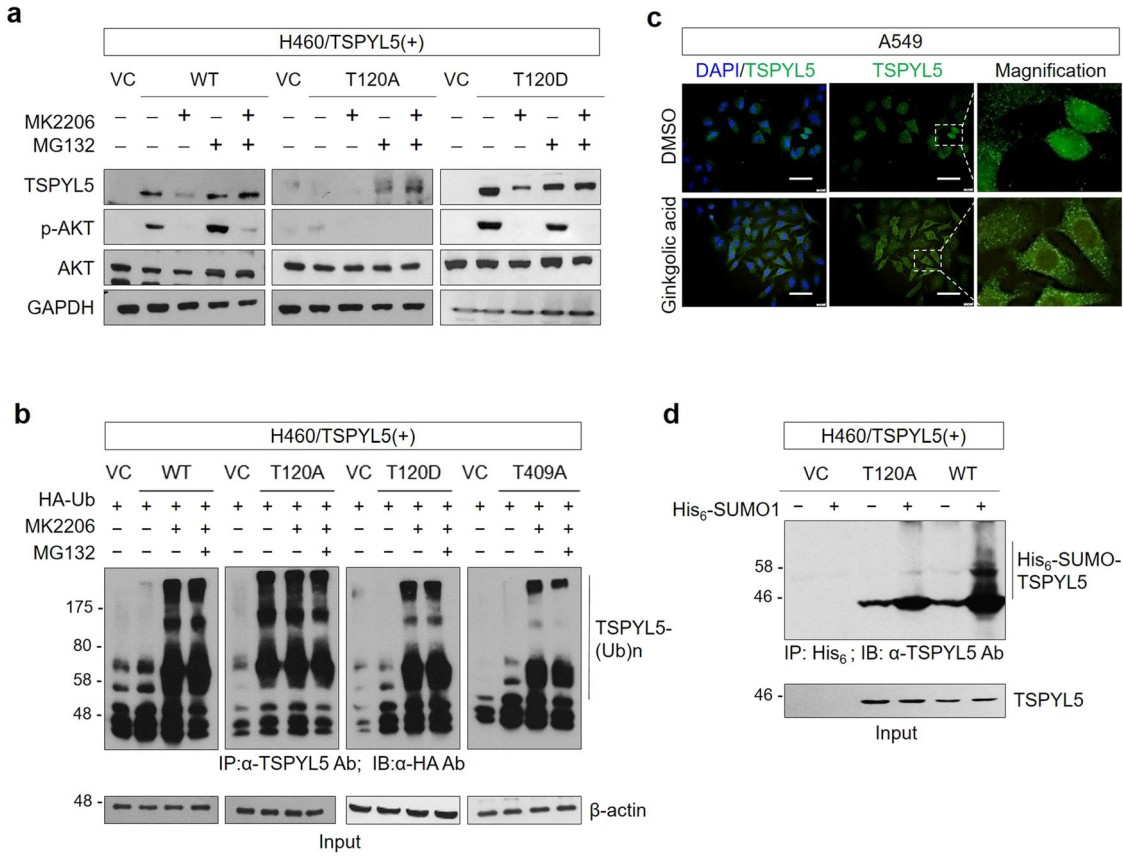

**Fig. 6 AKT-mediated phosphorylation of threonine-120 in TSPYL5 blocks ubiquitination and induces SUMOylation. a** The effect of AKT inhibitor (MK2206) or 26S proteasomal inhibitor MG132 on TSPYL5 expression in H460 cells transfected with TSPYL5 (WT, mutant T120A or T120D). VC: pcDNA3.1-transfected control. The activity of AKT inhibitor was confirmed by Western blot analysis of p-AKT. **b** In vivo ubiquitination assay of H460 cells transfected with HA ubiquitin and TSPYL5 or related mutants (T120A, T120D, and T409A) and treated with or without MK2206. Cells were lysed and immunoprecipitated with anti-TSPYL5 antibody. Ubiquitinated TSPYL5 was determined with anti-HA antibody. **c** Treatment of A549 cells with the SUMO1 modification inhibitor ginkgolic acid inhibited nuclear translocation of TSPYL5. A549 cells attached on coverslips were treated with ginkgolic acid (3 μM, 4 h), and intracellular staining of TSPYL5 was performed after fixation and permeabilization. Scale bar: 20 μm. **d** In vivo SUMOylation assay of H460 cells transfected with $His_6$-SUMO1 and TSPYL5 or related mutants T120A. Cell lysates were precipitated with Ni-agarose bead and Western blot analysis was performed with anti-TSPYL5 antibody.

into the nucleus the same as WT-TSPYL5 (Supplementary Fig. 5a and Fig. 5d), showed similar SUMO1 conjugation to that of WT-TSPYL5 (Supplementary Fig. 7). Collectively, we concluded that stabilized TSPYL5 via T120 phosphorylation receives a SUMOylation modification for its nuclear entry, thus promoting its function as a CSC-associated factor.

**Phosphorylated TSPYL5 at threonine-120 functions as a transcriptional regulator for ALDH1, CD44, and PTEN, which enhances cancer stem-like properties.** CSC-like properties of NSCLC cells depending on the expression and phosphorylation of TSPYL5 were confirmed again by TSPYL5 overexpression in ALDH1$^{low}$ or CD44$^{low}$ A549 cells. When ALDH1$^{low}$ A549 cells were transfected with WT-TSPYL5 or T120D-TSPYL5, the cells exhibited a restored sphere-formation capacity of up to 70–80% of that of the ALDH1$^{high}$ cells. However, overexpression of the T120A-TSPYL5 mutant did not enhance the sphere-formation capacity of the ALDH1$^{low}$ cells (Fig. 7a). Also, when WT-TSPYL5 was overexpressed in CD44$^{low}$ cells, the EMT-associated, CSC-like properties were enhanced (Fig. 7b). These results indicate that TSPYL5 plays an important role in maintaining CSC-like properties by regulating ALDH1 and CD44 in NSCLC cells.

ALDH1 and CD44 were also upregulated in H460 cells with overexpression of either WT-TSPYL5 or T120D-TSPYL5, as shown by FACS analysis using Aldefluor or CD44 staining (Fig. 7c, d, upper panels). The elevated expression levels of ALDH1 and CD44 in these cells were further confirmed by RT-PCR and Western blot analysis (Fig. 7c, d, lower panels). Overexpression of WT-TSPYL5 or the T120D-TSPYL5 mutant also inhibited the cellular expression of PTEN at both the transcriptional and protein levels (Fig. 7d, lower panels). In contrast, the T120A-TSPYL5 mutant did not affect the expression of ALDH1, CD44, or PTEN (Fig. 7c, d, lower panels).

As shown above (Figs. 5, 6), TSPYL5 is stabilized via phosphorylation by activated AKT, which blocks its ubiquitination and promotes its nuclear translocation. Moreover, phosphorylated TSPYL5 affects the expression of ALDH1, CD44, and PTEN at the transcriptional level. Therefore, we investigated whether TSPYL5 phosphorylation at threonine-120 is required for TSPYL5 to function as a transcriptional regulator of ALDH1, CD44, and PTEN by performing chromatin immunoprecipitation (ChIP) analyses (Fig. 7e and Supplementary Fig. 8). In A549 cells, the specific promoter regions of ALDH1, CD44, and PTEN were amplified from the immunoprecipitants that were collected using the anti-TSPYL5 antibody. Luciferase reporter assays also confirmed that WT-TSPYL5 and T120D-TSPYL5 function as positive transcriptional activators of CD44 and ALDH1 and as negative transcriptional regulators of PTEN (Fig. 7f). However,

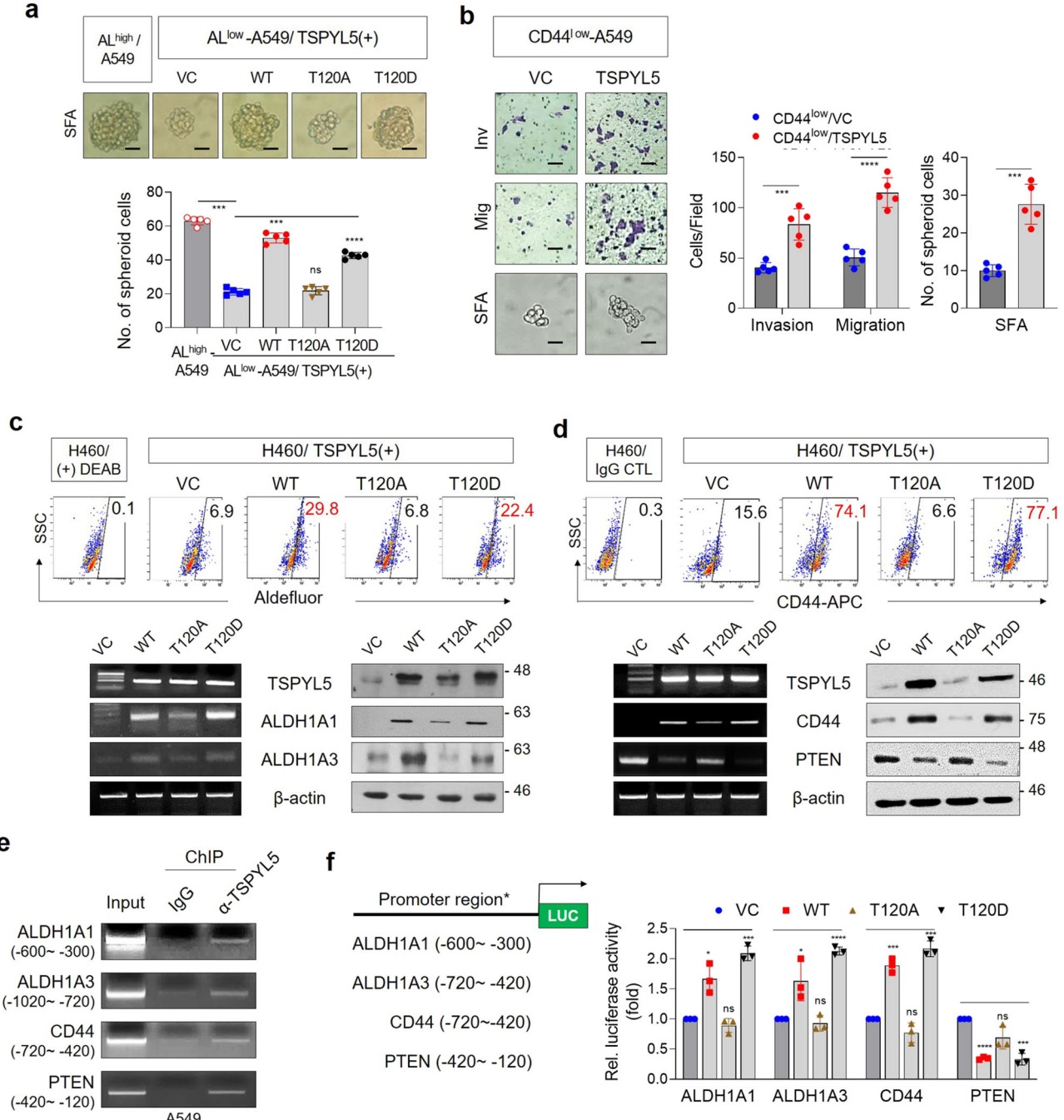

**Fig. 7 Phosphorylation of threonine-120 in TSPYL5 is critical for CSC-like characteristics of cancer cells by driving transcriptional activation of CSC-related factors. a** Sphere-formation assay of ALDH1$^{low}$ A549 cells transfected with WT or site-directed mutagenic TSPYL5 expression vectors (T120A and T120D). ALDH1$^{high}$ A549 cells were used as control. $n = 5$ independent experiments. Scale bar: 20 µm. **b** Invasion/migration and sphere-formation ability of CD44$^{low}$ cells following transfection with WT-TSPYL5 expression vector. $n = 5$ independent experiments. **c** ALDH1A1/A3 expression analyzed by flow cytometric analysis with Aldefluor stainining in H460 cells transfected with non mutagenic (WT: T120) or site-directed mutagenic TSPYL5-expression (T120A or T120D) vectors (upper panel). Sorted cells depending on Aldefluor staining were restained to use as gating controls. RT-PCR and Western blot analysis were also performed (lower panel). **d** CD44 expression analyzed by flow cytometric analysis with CD44-APC staining in H460 cells transfected with nonmutagenic (WT: T120) or site-directed mutagenic TSPYL5-expression (T120A or T120D) vectors (upper panel). Sorted cells depending on CD44-APC were re-stained to use as gating controls. RT-PCR and Western blot analysis of CD44 and PTEN were also performed (lower panel). **e** Chromatin immunoprecipitation (ChIP) assays with TSPYL5 antibody evaluating the effect of TSPYL5 on the transcriptional regulation of ALDH1, CD44, and PTEN in A549 cells. PCR for ChIP assay was performed using primers (Supplementary Table 7) positioned between −900 and +120 bp upstream of transcriptional start site of each gene (Supplementary Fig. 8) and representative results were shown. **f** Luciferase assay evaluating the effect of TSPYL5 on ALDH1, CD44, and PTEN expression in H460 cells transfected with WT or mutagenic TSPYL5-expression vectors (T120A and T120D). Promoter reporter vectors were constructed with specific promoter regions of each gene, which were amplified in ChIP assays as shown in the schematic representation. $n = 3$ independent experiments. Data represent mean ± s.d. using two-tailed *t*-test. NS not significant, $^{*}p < 0.05$, $^{**}p < 0.01$, $^{***}p < 0.001$, $^{****}p < 0.0001$.

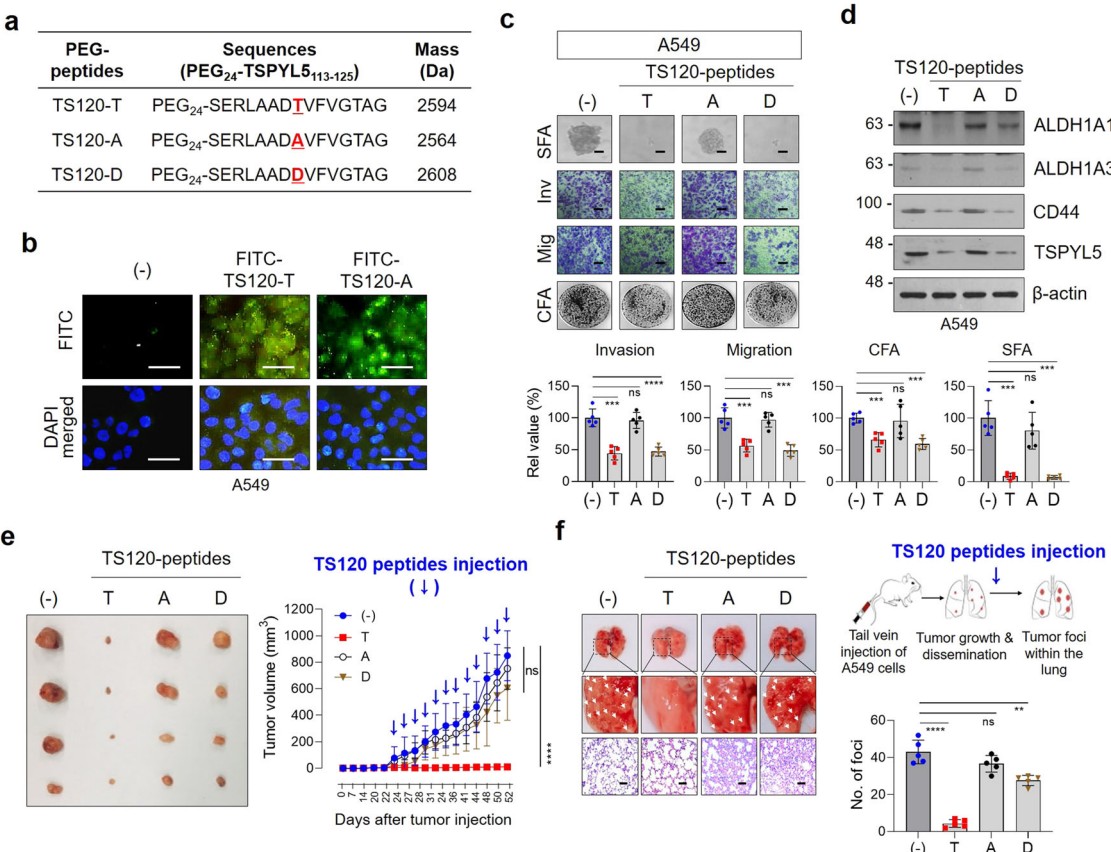

**Fig. 8 Cell-permeable TSPYL5 peptide can block EMT and tumorigenicity in in vitro and in vivo systems. a** Sequences of cell-permeable PEGylated TS120 peptides. **b** Cell permeability of FITC-labeled TS120 peptides (FITC-TS120-T and FITC-TS120-A) into the cells shown by fluorescence microscopy. A549 cells attached on coverslips were incubated with FITC-labeled TS120 peptides. **c** TS120 peptides affect metastatic capacity (invasion/migration), sphere-formation ability (SFA), and colony-formation ability (CFA) of A549 cells. $n = 5$ independent experiments. Scale bar: 20 µm. **d** Effects of TS120 peptides on the cellular level of TSPYL5, ALDH1, and CD44. Peptides were treated at a concentration of 10 µM for 12 h. **e** Effects of TS120 peptides on tumorigenic capacity of ALDH1$^{high}$-A549 cells in vivo. ALDH1$^{high}$ cells ($1 \times 10^6$) sorted from A549 cells were subcutaneously injected to athymic BALB/c female nude mice ($n = 4$ per group). After tumor formation (13.4 mm$^3$, 3 weeks after injection), TS120 peptides were intratumorally injected twelve times at 2-day intervals (one injection dose was 40 µg; total 480 µg). Tumor size was measured every two or three days, and the volume was calculated by shortest diameter$^2$ × longest diameter$^2$. Tumor burdens were removed from mice after 8 weeks of cell injections and photographed. **g** Effects of TS120 peptides on metastatic dissemination of ALDH1$^{high}$-A549 cells in vivo. Peptides were first administered at 4 days after intravenous injection of ALDH1$^{high}$-A549 cells into BALB/c nude mice (5 weeks of age, $n = 5$ per group). TS120 peptides were then treated twelve times every other day by intravenous injection (40 µg/dose/mouse; total 480 µg). The white arrow indicates the metastatic foci in the lung and the number of metastatic foci per lung was plotted. Representative images of H&E-stained lung sections show tumor foci. Data represent mean ± s.d. using two-tailed $t$-test. NS not significant, $^{**}p < 0.01$, $^{***}p < 0.001$, $^{****}p < 0.0001$.

the T120A-TSPYL5 mutant did not affect the transcription of these genes, implicating that TSPYL5 phosphorylation at threonine 120 is necessary for its function as a transcriptional regulator.

**TSPYL5-antagonistic peptides decrease TPSYL5 expression and consequently inhibit EMT-associated tumorigenicity in CSC-like NSCLC cells.** To further validate that TSPYL5 phosphorylation at threonine 120 plays a key role in the EMT-associated, CSC-like properties of NSCLC cells, we designed peptide inhibitors based on the TSPYL5 amino acid sequence to prevent threonine 120 phosphorylation by competitively binding to AKT (TS120 peptides: Fig. 8a). These peptides were PEGylated at their amino-termini to confer cell permeability and stability[54], and their intracellular delivery was confirmed by fluorescence imaging (Fig. 8b). Treatment of NSCLC A549 cells with the PEGylated TSPYL5-derived peptides, particularly TS120-T and TS120-D, resulted in significantly reduced self-renewal and metastatic capacities, as shown by sphere-forming assay and invasion/

migration assays (Fig. 8c). In contrast, TS120-A did not affect these cellular properties. Consistent with these results, TS120-T and TS120-D reduced the expression levels of ALDH1 and CD44, whereas TS120-A did not (Fig. 8d). Also, as expected, TS120-T and TS120-D peptides reduced TSPYL5 levels, which may be due to the competitive binding to AKT, thus resulting in decreased phosphorylation and stabilization of TSPYL5. Contrary to the effects observed in A549 cells, TS120-T peptides did not affect the self-renewal and metastatic capacities of H460 cells with a low level of endogenous TSPYL5 (Supplementary Figs. 9, 1d). These results suggest that TSPYL5 phosphorylation at threonine 120 is essential for EMT-associated CSC-like cell properties, and that PEGylated TS120-T or -D peptides can be used as inhibitory molecules of CSC-like cells with high TSPYL5 expression. The antagonistic effects of the TSPYL5 sequence-derived peptides were also investigated in vivo. Mice that were inoculated with ALDH1$^{high}$ A549 cells showed a gradual increase in tumor burden, which reached a volume of approximately 1,000 mm$^3$ at day 52 after inoculation (Fig. 8e). However, when the TS120-T

peptides were intratumorally injected every 3 days after the tumor volume reached approximately 13.4 mm³, the tumor growth dramatically regressed. On the contrary, TS120-A peptides did not significantly affect tumor growth. Additionally, in a metastatic lung cancer model that was established by intravenous injection of ALDH1^high A549 cells into nude mice, TS120-T peptides markedly inhibited metastatic dissemination of the cancer cells to the lungs (Fig. 8f); however, TS120-A peptides did not. Interestingly, despite the inhibitory effects of the TS120-D peptides in cellular assays (Fig. 8c, d), significant inhibitory effects were not observed for in vivo models (Fig. 8e, f), which may be due to certain effects on peptides that only occur in vivo, such as proteolytic attack. Immunohistochemical analysis of tumor tissues from the in vivo models confirmed that TS120-T peptide treatment reduced the cellular levels of TSPYL5, thereby resulting in decreased CD44 and increased PTEN levels (Supplementary Fig. 10). As expected, treatment with TS120-A or TS120-D peptides did not alter the levels of TSPYL5, CD44, or PTEN in tumor cells from the in vivo models.

**TSPYL5-antagonistic peptides sensitize NSCLCs to γ-radiation and EGFR tyrosine kinase inhibitors**. Finally, we investigated whether blocking the phosphorylation of TSPYL5 at threonine-120 could overcome the acquired resistance of NSCLC cells to γ-radiation or EGFR TKIs. Treatment of TSPYL5-expressing A549 cells with TS120-T or TS120-D peptides significantly increased the sensitivity of the cells to γ-radiation as shown by colony-forming assay, whereas TS120-A peptides did not (Fig. 9a). TS120-T peptide also inhibited the self-renewal and metastatic properties of the radiation-resistant A549 cells (Fig. 9b). We also examined whether targeting TSPYL5 phosphorylation could reverse the drug resistance of the NSCLC cells. Targeting TSPYL5 with TS120-T peptide improved the sensitivity of TSPYL5-expressing A549 cells to EGFR TKI gefitinib (Fig. 9c), which showed similar result with TSPYL5-knockdown cells (Supplementary Fig. 11a). Similarly, TS120-T peptide treatment inhibited the CSC-like properties and tumorigenicity of the gefitinib-resistant H460 cells (Fig. 9d). Treatment with the TS120-T peptides in other lung cancer cells with high levels of TSPYL5, such as H358 and H2009 (Supplementary Fig. 1), also inhibited cell growth and enhanced the sensitivity of the cells to γ-radiation and gefitinib (Supplementary Fig. 11b).

In conclusion, these results suggest that phosphorylation of TSPYL5 at threonine-120 is essential for eliciting cancer stemness and inherent or acquired resistance of NSCLC cells to radiation or targeted drugs, and that TS120-T peptides may be used to sensitize CSC-like NSCLC cells to these therapies. We have studied TSPYL5 as CSC-associated factor mainly in lung CSCs; however, its expression is not restricted to lung cancer. TSPYL5 has been confirmed as a cancer stemness factor also in other types of cancer, including liver cancer and pancreatic cancer (Supplementary Fig. 12a–d), thus warranting further studies on TSPYL5 and AKT activation in other tumors.

## Discussion

TSPYL5 is expressed at low levels in most human adult tissues, except the testis, and gene silencing of TSPYL5 is mediated by promoter DNA methylation[55]. However, TSPYL5 expression is often associated with poor prognosis of cancers, in which promoter DNA methylation is erased[56]. The elevated TSPYL5 can cause the poor clinical outcomes by suppressing p53 levels by inhibiting deubiquitinase USP7[23,24]. TSPYL5 also contributes to the survival of ALT-positive cancer cells via inhibition of the USP7/POT1 E3 ligase complex[25]. Unlike the function of TSPYL5 as a USP7 regulator, our study had revealed that TSPYL5 is

associated with aberrant AKT activation in drug- and radiation-resistant NSCLC cells[27]. And here, we revealed that AKT-mediated phosphorylation of TSPYL5 represses the transcription of PTEN and maintains persistently activated AKT via a positive-feedback loop between the AKT/TSPYL5/PTEN and PTEN/PI3K/AKT signaling pathways. Furthermore, we found that phosphorylated TSPYL5 acts as a transcriptional activator of CSC-associated factors, ALDH1 and CD44, which lead to the sustained self-renewal ability of CSCs (Fig. 10).

AKT has many well-known, crucial roles in proliferation, anti-apoptosis, protein synthesis, and metabolism. Recently, AKT has also been suggested to have regulatory significance in cancer stemness[57] because the AKT-dependent phosphorylation pattern has been identified within the key stemness-inducing transcription factors, OCT-4, SOX-2, and KLF4[58,59]. Moreover, additional modification patterns, including phosphorylation-dependent inhibition of ubiquitination and SUMOylation, have been identified for these stemness factors, implicating that overall control on stemness-related transcription factors by AKT-dependent regulatory mechanisms determines the cellular fate of stem cells.

Interestingly, these stemness factors are not only stabilized and activated by AKT, but they also regulate AKT activity in a positive feedback manner, resulting in the maintenance of self-renewal potential without any external activation signals. AKT-activated SOX-2 and KLF4 cooperatively drive PIK3CA expression, thus enhancing PI3K/AKT activity and tumorigenesis[60]. NANOG, another key stemness factor regulated by the PI3K/AKT pathway[61], enhances the transcription of TCL1a, an AKT activator protein[62], thus creating a positive-feedback loop for AKT activation. The regulatory mechanism of TSPYL5 is similar to that of stemness factor Nanog, SOX-2, or KLF-4: TSPYL5 is regulated by AKT-dependent phosphorylation, which stabilizes TSPYL5 and also induces SUMOylation-dependent nuclear translocation. Additionally, TSPYL5 functions as an AKT activator via suppression of PTEN expression. Altogether, we conclude that stemness factors, including SOX2, KLF4, NANOG, and TSPYL5, function to reinforce AKT signaling in CSCs via similar feedback signaling loops.

More importantly, the function of TSPYL5 as a CSC-associated factor is relevant to its role as a transcription activator of CD44 and ALDH1. CD44 and ALDH1 are well-known cancer stemness drivers; however, their expression has not been studied in detail. Only a few factors, including STAT3, β-catenin, and NF-kB, have been reported as their transcriptional regulators[63–65]. Here, we suggest TSPYL5 as a novel cancer stemness factor that simultaneously activates ALDH1 and CD44 transcription.

In clinical trials, CSC-associated signaling pathways have been attractive targets to overcome cancer cell resistance against conventional therapies. However, CSC signaling pathways, including WNT, Hedgehog, and NOTCH, are upstream targets in signaling pathways, and function similarly in normal adult stem cells. Therefore, targeting such pathways has been unsuccessful for cancer therapy[66]. To improve the efficiency of targeting CSCs, more sophisticated knowledge about CSC-specific genes, and the underlying mechanisms for maintenance of CSC characteristics are needed. Also, novel strategies for targeting conventional undruggable targets must be developed. Recently, SOX-2 has been suggested as an attractive anticancer target[67,68]. Although it has high relevance to cancer progression and drug resistance, directly targeting SOX-2 had been difficult because transcription factors are regarded as undruggable targets. Nonetheless, studies on the regulation of SOX-2 expression revealed that a neddylation inhibitor, MLN4924, can be used to target SOX-2 for anti-cancer therapy, which represses transcription of SOX-2. The self-renewal ability via the aberrant signaling of the AKT/TSPYL5/PTEN positive-feedback loop shown in this study is a remarkable

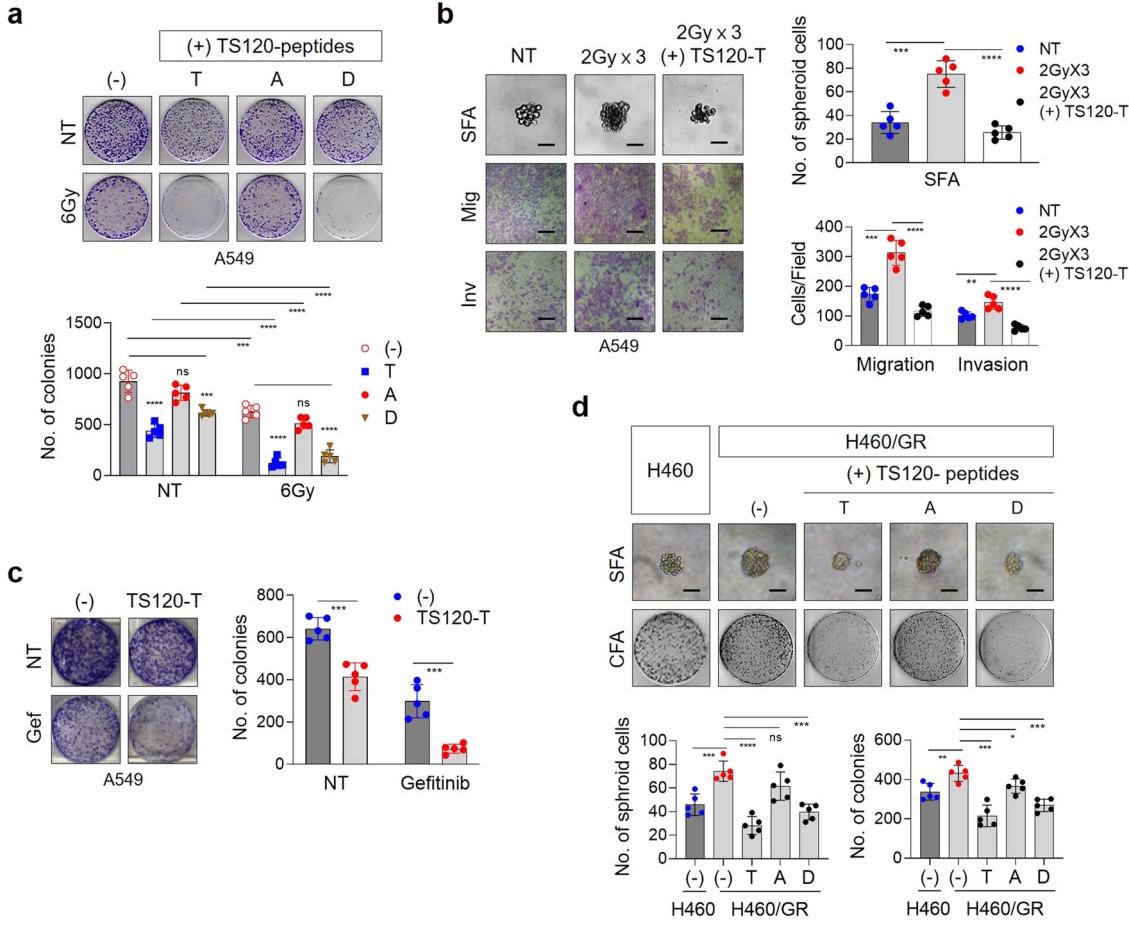

**Fig. 9 TSPYL5-antagonistic peptides reduce the therapeutic resistance of CSC-like NSCLC cells. a** Radio-sensitizing effect of TS120 peptides on A549 cells. Cells were treated with TS-120 peptides (10 μM) and irradiated with a total dose of 6 Gy using $^{60}$Co γ-ray at a dose rate of 0.2 Gy/min. Cell survival after irradiation was determined by colony-forming assay. $n = 5$ independent experiments. **b** Loss of CSC-like properties of γ-irradiated A549 cells by TS120-T peptide. A549 cells, which were treated with TS120-T peptide (10 μM), were irradiated with fractionated exposures (2 Gy × 3, total 6 Gy). Self-renewal potential and EMT properties were examined by sphere-forming and invasion/migration assays. $n = 5$ independent experiments. Scale bar: 20 μm. **c** Drug-sensitizing effect of TS120 peptides on A549 cells. Gefitinib (1 μM) and TS120-T peptide (10 μM) were treated on A549 cells and cell survival was examined by colony-formation assay. $n = 5$ independent experiments. **d** Inhibition of self-renewal and colony-formation ability of H460 cells enhanced by gefitinib resistance (GR) by TS120 peptides. TS120 peptides (10 μM) were treated on GR-H460 cells and cell survival and self-renewal ability were determined. $n = 5$ independent experiments for each group. Data represent mean ± s.d. using a two-tailed $t$-test. NS not significant, $^{*}p < 0.05$, $^{**}p < 0.01$, $^{***}p < 0.001$, $^{****}p < 0.0001$.

property of CSCs; however, it can be disrupted by eliminating TSPYL5. Targeting TSPYL5 has to be performed in a specific manner without disturbing the PI3K/AKT cellular signaling network in healthy cells. In this regard, we had designed antagonist peptides (TS120-T peptides) that target previously deemed undruggable target TSPYL5 via the inhibition of its AKT-dependent phosphorylation. The cell-permeable PEGylated TS120-T peptides specifically induced TSPYL5 degradation and targeted TSPYL5-expressing cancer cells. TSPYL5 degradation resulted in disassembling of aberrant AKT/TSPYL5/PTEN cyclic signaling as well as sequential suppression of cancer stemness properties and therapeutic resistance of CSCs.

Despite intrinsic drawbacks, including in vivo instability and membrane permeability, modified peptides have been developed as drug candidates to disrupt protein-protein interactions[69] and to target or inhibit intracellular molecules, such as receptor tyrosine kinases[70]. We expect PEGylated TS120-T peptides or their derivatives, which destabilize TSPYL5 in CSCs, to be potential candidates for the development of CSC therapies. Additionally, protein destabilization of other stemness factors, such as SOX-2 or OCT-4, by mimicking their AKT-dependent phosphorylation

domains, may also be a practical approach to improve cancer treatment efficacy and warrants further investigation.

## Methods

**Cell culture and irradiation**. Cell lines were purchased from the American Type Culture Collection (ATCC, Manassas, VA, USA) and were cultured in RPMI-1640 or DMEM media (Invitrogen, Carlsbad, CA, USA) containing 1% penicillin/streptomycin and 10% fetal bovine serum (HyClone, Logan, UT, USA). Cells were incubated at 37 °C in a humid atmosphere with 5% $CO_2$ and 95% air. For irradiation experiments, cells were seeded into a T-25 flask at a density of $1 \times 10^5$ cells/5 mL, incubated for 1 day, and then exposed by $^{60}$Co γ-ray at the indicated dose (dose rate: 0.2 Gy/min). For fractionated γ-radiation, cells were irradiated 3 or 10 times at a dose of 1 or 2 Gy at 3-day intervals. The exposed cells were harvested the next day and replated for further analysis. Highly gefitinib-resistant (GR) H460 cells were generated by the stepwise escalation of concentration of gefitinib from 1 to 10 μmol/L over 4 months.

**DNA constructs and mutagenesis**. Complementary DNAs encoding full-length human TSPYL5, ALDH1A1, ALDH1A3, AKT1, and PTEN were prepared by RT-PCR using total RNA isolated from H460 or A549 cells. The PCR primers that were used are listed in Supplementary Table 2. The i-GCapture solution (iNtRON Biotechnology, Seongnam, Korea) was added to the PCR mixture to amplify the GC-rich TSPYL5 template. Each cDNA insert was cloned into the HindIII and EcoRI sites of pcDNA3.1 (Invitrogen). The alanine-substituted TSPYL5 mutant

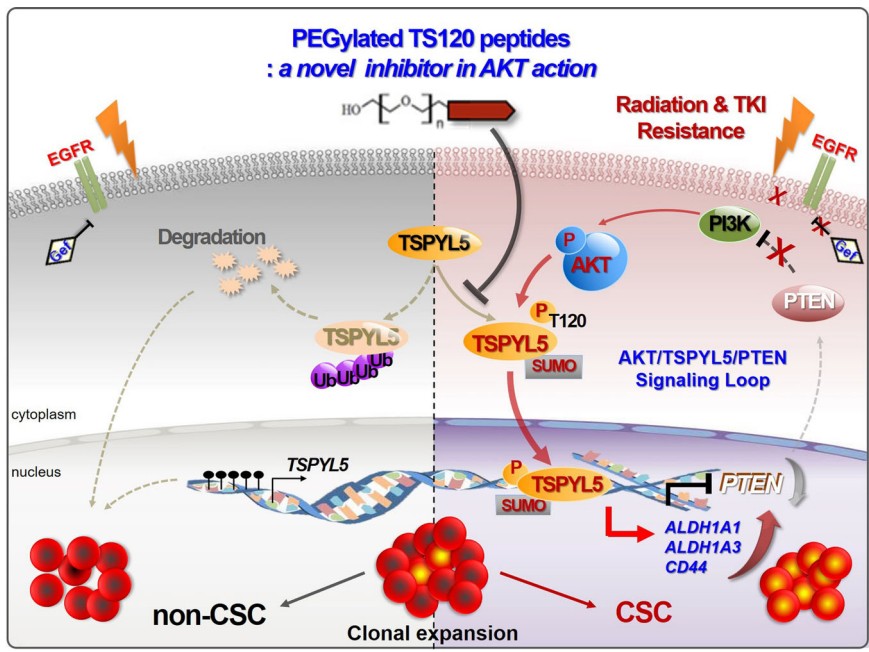

**Fig. 10 Schematic model of CSC-associated AKT/TSPYL5/PTEN loop signaling pathway.** TSPYL5-mediated signaling pathway, which regulates the expression of ALDH1, CD44, and PTEN, enforces cancer stemness in a positive feedback manner. It is initiated by AKT-dependent phosphorylation of TSPYL5, followed by suppression of PTEN expression and consequent activation of AKT. TSPYL5 is a critical component of the CSC-associated loop signaling pathway. On the other hand, TSPYL5-depleted cells by promoter methylation or protein degradation are hard to maintain cancer stemness, resulting in high sensitivity to radio- or chemotherapy. Therefore, the strategies for TSPYL5 depletion, which can disrupt the self-renewal program of CSCs, can be novel methods for cancer treatment.

(T120A-TSPYL5, T120D-TSPYL5, T177A-TSPYL5, T326A-TSPYL5, and T409A-TSPYL5) expression vectors were generated using QuikChange Multi Site-Directed Mutagenesis Kits (Agilent Technologies, Santa Clara, CA, USA; mutagenesis primer sequences are listed in Supplementary Table 3).

**Chemicals and antibodies.** All chemicals for the experiments were reagent grade or better. The AKT inhibitor, MK2206, was obtained from Santa Cruz Biotechnology (Santa Cruz, CA, USA). The PI3K inhibitor, LY294002, was obtained from Cell Signaling Technology (Beverly, MA, USA). The SUMOylation inhibitor, ginkgolic acid, was purchased from Abcam (Cambridge, MA, USA). The protease inhibitor, MG132, was provided by Millipore (Billerica, MA, USA). Antibodies used in this study are listed in Supplementary Table 4. CD44 was detected with pan-CD44 antibodies [CD44-APC, CD44 (8E2)], which detect all CD44 isoforms (CD44s and various CD44v).

**Western blot and immunoprecipitation assays.** Cells were lysed in TX100 lysis buffer [20 mM Tris-HCl (pH 7.5) buffer containing 150 mM NaCl, 1 mM EGTA, 1 mM EDTA, and 0.5% Triton X-100] with a protease inhibitor cocktail (Sigma-Aldrich, St Louis, MO, USA). Protein concentration was measured using Bradford reagent (Bio-Rad, Hercules, CA, USA). For western blot analysis, equal amounts of protein were separated in 10% sodium dodecyl sulfate (SDS)-polyacrylamide gels and then transferred to Hybond nitrocellulose membranes (Amersham Pharmacia, Pittsburgh, PA, USA). Gels were blocked for 1 h at room temperature in phosphate-buffered saline (PBS) containing nonfat milk (10%) and Tween-20 (0.1%). Membranes were treated with the respective antibodies overnight in a cold chamber. After washing with tris-buffered saline, membranes were treated with a horseradish peroxidase-labeled secondary antibody (Abcam) and visualized using the West-Zol Enhanced Chemiluminescence Detection Kit (iNtRON Biotechnology). Immunoprecipitation was performed overnight at 4 °C using 2 mg of cell lysates with the appropriate amount of the specific antibodies and protein A/G Ultralink Resin (Invitrogen). After washing with TX100 lysis buffer, immunoprecipitants were resuspended in 2× SDS sample buffer, separated in gels, and analyzed by western blot using the respective antibodies.

**Silencing RNA (siRNA) targeting.** A549 or H460 cells were transfected with siRNA targeting the TSPYL5 (Invitrogen), PTEN, AKT, ALDH1A1, and ALDH1A3 (Bioneer, Daejeon, Korea) genes (siRNA sequences are listed in Supplementary Table 5) using Lipofectamine RNAi MAX reagent (Invitrogen). Stealth RNAi Negative Control Medium GC (Invitrogen) was used as a negative control. Cells were cultured for 72 h after transfection and gene expression levels were measured by RT-PCR.

**RT-PCR.** TRIzol reagent (Invitrogen) was used for total RNA extraction from cancer cells under various conditions. First-strand cDNA was prepared from 1 μg of extracted RNA using oligo-dT primers and a cDNA synthesis kit (iNtRON Biotechnology). Prepared cDNA served as templates for PCR amplification with specific primers (Supplementary Table 6). PCRs were performed under the following conditions: initial denaturation at 94 °C for 5 min, 25−30 cycles at 94 °C for 1 min, 56 °C for 1 min, and 72 °C for 90 s; and a final extension at 72 °C for 10 min. The amplified PCR products were separated in 1% agarose gels and visualized by EcoDye (Biofact, Daejeon, Korea).

**Invasion and migration assays.** Invasion and migration assays were performed as previously described64 using Matrigel-coated chambers (8-μm pores; BD Biosciences, San Jose, CA, USA). After staining, cells were counted using a light microscope in four randomly selected fields.

**Sphere-formation assay.** Cells were placed in stem cell-permissive Dulbecco's Modified Eagle Medium (DMEM-F12, Invitrogen) containing epidermal growth factor (20 ng/mL), basic fibroblast growth factor (20 ng/mL), and B27 serum-free supplement (Invitrogen). Suspended cells were seeded into ultra-low-attachment 96-well plates (Corning, Inc., Corning, NY, USA) at a density of 1 or 2 cells/well and incubated at 37 °C in a 5% CO$_2$-humidified incubator. The next day, each well was visually checked for the presence of a single cell. After 10–14 days, spheres were quantitated using inverted phase-contrast microscopy and photographed.

**Colony-forming assay.** Cells were seeded into 35-mm culture dishes at a density of 1,000 cells per plate and allowed to attach overnight. The next day, cells were exposed to a 2-Gy dose of γ-radiation. After 10–14 days, cells were stained with 0.5% crystal violet, and colonies (groups of ≥50 cells) were counted. Clonogenic survival was expressed as a percentage relative to the nonirradiated controls.

**Cancer stem cell (CSC) sorting from the A549 NSCLC cell line.** Aldefluor assays (STEMCELL Technologies, Vancouver, BC, Canada) were performed to isolate and characterize CSC populations from NSCLC cell cultures according to the manufacturer's instructions. Cells ($1 \times 10^6$) were harvested from cell cultures and resuspended in Aldefluor assay buffer containing aldehyde dehydrogenase (ALDH) substrate. As a negative control, aliquots of Aldefluor-exposed cells were immediately quenched with the specific ALDH inhibitor, N,N-diethylamino-benzaldehyde (DEAB). After 30 min of incubation at 37 °C, the cells were washed and sorted as either ALDH1$^{high}$ or ALDH1$^{low}$ cells using a FACSAria cell sorter (BD Biosciences).

**Fluorescence microscopy**. Cells ($5 \times 10^4$) were grown on glass coverslips in 6-well plates and fixed with 4% paraformaldehyde. Following cell fixation, cells were incubated with anti-TSPYL5 antibody in a permeabilization solution of PBS containing 0.1% Triton X-100 and 1% bovine serum albumin at 4 °C overnight. Staining was visualized using an Alexa Fluor 488-conjugated anti-rabbit IgG antibody (Invitrogen). Cell permeability of TS-120 peptide was estimated using FITC-labeled TS120 peptides. Cells grown on glass coverslips were treated with FITC-labeled TS120 peptides (TS120-T or TS120-A) at 10 μM overnight. Nuclei were counterstained using 4,6-diamidino-2-phenylindole (DAPI, Sigma-Aldrich). Stained cells were analyzed using a Zeiss LSM510 Meta microscope (Carl Zeiss MicroImaging GmbH, Göttingen, Germany).

**Subcellular fractionation**. After washing and scraping in PBS, cells were pelleted by centrifugation for 5 min at $200 \times g$ at 4 °C and incubated in a hypotonic buffer (1.5 mM MgCl$_2$, 10 mM KCl, and 0.1 mM EGTA in 10 mM HEPES, pH 7.2) for 30 min at 4 °C with shaking. Cells were sonicated and disrupted, and then they were centrifuged for 10 min at $720 \times g$ at 4 °C to pellet the nuclei. Nuclear pellets were resuspended in nuclear lysis buffer (5 mM EDTA, 150 mM NaCl, and 1% Triton X-100 in 10 mM Tris-HCl, pH 7.5), incubated for 1 min in an ultrasonic bath, and then incubated for 30 min at 4 °C with shaking. The nuclei-free supernatant was subjected to a second centrifugation ($10,000 \times g$) for 45 min at 4 °C to separate the membrane (pellet) from the cytoplasmic (supernatant) fractions. Total cytoplasmic and nuclear proteins were analyzed by western blot.

**In vivo ubiquitination assay**. H460 cells were transfected with plasmid encoding HA-tagged ubiquitin (Ub) purchased from Addgene (#17608) as well as other plasmids expressing the TSPYL5 derivatives. After 48 h, cells were treated with 20 μM MK2206 (Santa Cruz Biotechnology) or MG132 for 1 h. Cells were harvested and lysed using NP-40 lysis buffer (150 mM NaCl, 0.5% Nonidet P-40, and 1.5 mM EDTA in 50 mM Tris-HCl, pH 8.0) containing 2 mM N-ethylmaleimide and protease inhibitor cocktail. To denature the proteins, cell lysates were incubated at 65 °C for 20 min with 1 mM dithiothreitol and 1% SDS. Western blot analysis was performed with specific antibodies using total cell extracts or a further immunoprecipitated pellet using specific or HA antibodies.

**In vivo SUMOylation assay**. To generate and purify His-tagged SUMO1 conjugates in vivo, H460 cells were transfected with plasmids encoding His-SUMO1 (Addgene, #17271) and pcDNA3.1-TSPYL5/WT or pcDNA3.1-TSPYL5/T120A. Cells were harvested after 36 h, and 5% of the cells were dispersed in NP-40 lysis buffer and analyzed by western blot as the input control. The remaining cells were lysed in buffer A (10 mM β-mercaptoethanol and 5 mM imidazole in 6 M guanidinium-HCl, pH 8.0). Lysates in buffer A were mixed with 20 μL of Ni-NTA-agarose beads (Qiagen, Venlo, Netherlands) and incubated at 4 °C overnight. Beads were washed sequentially with buffer A, buffer B (0.1 M Na$_2$HPO$_4$/NaH$_2$PO$_4$, 10 mM β-mercaptoethanol, and 8 M urea in 0.01 M Tris-HCl, pH 8.0), and buffer C (same as buffer B, except pH 6.3). Washed beads were incubated with 40 μL of elution buffer (30% glycerol, 0.72 M β-mercaptoethanol, 200 mM imidazole, and 5% SDS in 0.15 M Tris-HCl, pH 6.7) at room temperature. Eluates were analyzed by western blot.

**Chromatin immunoprecipitation assay**. ChIP assays were performed using EZ ChIP kits (Millipore) according to the manufacturer's instructions with minor modifications. Cells ($2 \times 10^6$ cells) were sonicated with SDS lysis buffer, and cell lysates were diluted with ChIP dilution buffer. After nonspecific binding was performed using protein A/G agarose beads, supernatants were incubated overnight with TSPYL5 antibody at 4 °C, and then fresh protein A/G agarose beads were added. After washing the pellet with washing buffers, samples were fractioned by elution buffer (0.1 M NaHCO$_3$ and 1% SDS) and sequentially treated with 5 M NaCl solution at 65 °C and 1 M Tris-HCl buffer (pH 6.5) containing proteinase K (10 mg/mL) and 0.5 M EDTA at 45 °C. After filtration of this sample, the eluate was used as the PCR template with promoter primers (Supplementary Table 7). PCR products were separated in 1% agarose gels and visualized by EcoDye (Biofact).

**Luciferase reporter assay**. Cells were seeded into a 24-well plate, and plasmid transfection was performed the next with Lipofectamine (Invitrogen) according to the manufacturer's protocol. Transfected plasmids were pGL4 luciferase reporter vectors encoding the target gene promoter regions, Renilla luciferase control reporter vector (pRL; Promega, Madison, WI, USA), and plasmids encoding TSPYL5 and/or its mutant forms or the corresponding empty control vector. Cells were collected after 72 h and luciferase activity was measured using the Dual-Luciferase Reporter Assay System (Promega) according to the manufacturer's instruction. Expression values of firefly luciferase were normalized to those of Renilla luciferase. For the construction of the luciferase reporter vectors, promoter region DNA of each gene, which was chromatin-immunoprecipitated with anti-TSPYL5 antibody, was amplified by PCR (primer set: ALDH1A1-3, ALDH1A3-4, CD44-3, and PTEN-2 in Supplementary Table 7) and inserted into the pGL4 luciferase reporter vector.

**Generation of phospho-120T/TSPYL5-specific antibody**. To generate an anti-phospho-120T/TSPYL5 antibody to discriminate between unmodified T120 and phospho-T120, rabbits were immunized with a TSPYL5-derived peptide phosphorylated at T120 as bovine serum albumin (BSA) conjugates. Synthesis of the TSPYL5 peptide containing T120 (SERLAAD**T**VFVGTAG: 113–127 amino acid residues of TSPYL5) and its phosphorylation was performed at the Korea Basic Science Institute (KBSI, Ochang, Korea). Immunization and antibody preparation were performed by AbFrontier (Seoul, Korea).

**Synthesis of cell-permeable TS120 peptides and treatment**. Cell-permeable PEGylated peptides containing the TSPYL5 sequence (TS120-T) and its derivatives (TS120-A or -D) were prepared by linking methoxy polyethylene glycol succinic acid (~2 kD) to the amino (N)-termini of 15-mer peptides derived from TSPYL5. Peptides were labeled with fluorescein-5-isothiocyanate (FITC) at the additional lysine residue at the N-termini of the TS120 peptides. The following peptide sequences were used: PEG-**Lys**(FITC)-SERLAADTVFVGTAG (FITC-TS120-T), PEG-**Lys**(FITC)-SERLAADAVFVGTAG (FITC-TS120-A). These peptides were synthesized at the Korea Basic Science Institute (Ochang, Korea) and confirmed by mass spectrometry. All peptides were solubilized to a concentration of 1 mM in 10% DMSO, and cells were treated at a concentration of 10 μM.

**Animal experiments**. Xenograft tumors were generated by subcutaneous injection of A549 lung cancer cells ($1 \times 10^6$ cells/100 μL) into the right flanks of BALB/c female nude mice (5 weeks of age, $n = 20$). The mice were randomly divided into four groups and intratumorally injected 12 times with the TS120 peptides (40 μg of PEGylated peptide/100 μL of 10% DMSO/mouse) every other day. Control mice were injected with an equal volume of 10% DMSO. Tumor size was measured with a caliper (calculated volume = shortest diameter$^2$ × longest diameter$^2$) at 2- or 3-day intervals. To establish the lung metastasis mouse model, A549 cells ($1 \times 10^6$ cells) were administered intravenously to BALB/c nude mice (5 weeks of age, $n = 5$ per group). TS120 peptides were then administered 12 times every other day by intravenous injection (40 μg/mouse). The methods were performed in accordance with relevant guidelines and regulations and approved by the Animal Care and Use Committee (IACUC) of the Korea Research Institute of Bioscience & Biotechnology (Approval No: KRIBB-AEC-19162).

**Statistics and reproducibility**. All experiments were performed with three or more biological replicates. Statistical analyses were performed using PRISM version 7.0 (GraphPad Software, San Diego, CA, USA). Experimental data are presented as the mean ± standard deviation (s.d.) of five or more independent experiments. Each exact $n$ value is indicated in the corresponding figure legend. Comparisons were performed using two-tailed paired Student's $t$-tests or one-way ANOVA. Not significant (ns): $P > 0.05$; significant: *$P < 0.05$; **$P < 0.01$; ***$P < 0.001$, and ****$P < 0.0001$, as indicated in individual figures.

**Reporting summary**. Further information on research design is available in the Nature Research Reporting Summary linked to this article.

## Data availability

Source data generated or analyzed during this study are provided in Supplementary Data 1 (source data for graphs) and a Supplementary information file (Supplementary Figs. 1–12, Supplementary Tables 1–7 and uncropped western blot images), but additional details can be obtained from the corresponding author on reasonable request.

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

## Acknowledgements
This work was supported by grants from the Ministry of Science and ICT (Nuclear Research and Development Program: NRF-2013M2A2A7043660) of the Republic of Korea and the Korea Research Institute of Bioscience & Biotechnology (KRIBB) Research Initiative Program.

## Author contributions
I.G.K. and E.W.C. conceived, designed, and led all the experiments; I.G.K. and E.W.C. prepared the paper; J.H.L., S.Y.K., C.K.H., R.K.K., and I.G.K. mainly performed the experiments; experiments related to radiation were performed by J.H.L. and R.K.K.; J.H.L. and E.W.C. arranged data.

## Competing interests
The authors declare no competing interests.
