## [Peer Review File · Communications Biology]

Reviewers' comments:

Reviewer #1 (Remarks to the Author):

Comments for the manuscript COMMSBIO-20-2273

This manuscript entitled "Targeting therapy-resistant lung cancer stem cells via disruption of the AKT/TSPYL5/PTEN positive-feedback loop" described the possible identification of testis-specific Y-like protein 5 (TSPYL5) as a CSC-associated factor that promotes stemness and epithelial-to-mesenchymal transition in therapy-resistant non-small cell lung cancer (NSCLC) cells.

Beside the limitation regarding the fact that this is developed in cancer cells lines and not in human samples, this manuscript is well-written and methodically well-organised.

• I have some comments for the Authors that need to be addressed:

1. You used ALDH1A1 and ALDH1A3 which are the most common isoforms for targeting cancer stem cells. Then you also used the biomarker CD44 in your manuscript as one of the most used targets for cancer stem cells.

I believe that this point needs to be better described and defined, because CD44 is made by several isoforms as well, that you did not described into your manuscript.

Suggested references:

Ranjeeta Thapa and George D. Wilson. The Importance of CD44 as a Stem Cell Biomarker and Therapeutic Target in Cancer. *Stem Cells International* Volume 2016, Article ID 2087204, 15 pages <http://dx.doi.org/10.1155/2016/2087204>

Hu B, Ma Y, Yang Y, Zhang L, Han H and Chen J: CD44 promotes cell proliferation in non-small cell lung cancer. *Oncol Lett* 15: 5627-5633, 2018

2. There are a recent manuscript which I would suggest to use, published in 2020 by Masciale V. et al in *Oncotarget* which demonstrated in human NSCLC samples that CD44 and ALDH do not identify the same population of cancer stem cells. Infact, they found that ALDH^{high} cells do not correlate with CD44⁺ cells but with CD44⁺/EpCAM⁺ cells. That is a very important concept, because you could also consider this aspect into your manuscript.

CD44⁺/EPCAM⁺ cells detect a subpopulation of ALDH^{high} cells in human non-small cell lung cancer: A chance for targeting cancer stem cells?

Valentina Masciale, Giulia Grisendi, Federico Banchelli, Roberto D'Amico, Antonino Maiorana, Pamela Sighinolfi, Alessandro Stefani, Uliano Morandi, Massimo Dominici, Beatrice Aramini *Oncotarget*. 2020 Apr 28; 11(17): 1545–1555. Published online 2020 Apr 28. doi: 10.18632/oncotarget.27568

3. Did you also consider the isoforms in CD44? If yes, which isoforms? The isoforms consideration is very important for the fact that, despite of ALDH marker, "CD44 is also expressed on healthy cells, making it difficult to be used to specifically differentiate CSCs" (Alhabbab RY).

Please consider these references :

Alhabbab RY. Targeting Cancer Stem Cells by Genetically Engineered Chimeric Antigen Receptor T Cells. *Front Genet.* 2020;11:312. Published 2020 Apr 22. doi:10.3389/fgene.2020.00312

Chen C, Zhao S, Karnad A, Freeman JW. The biology and role of CD44 in cancer progression: therapeutic implications. *J Hematol Oncol.* 2018;11(1):64. Published 2018 May 10. doi:10.1186/s13045-018-0605-5

4. With regards of the methods, it needs that you better specify which are the histopathological subtypes of NSCLC cells lines you used into your experiments. That is better for the reader, as well as also for giving the suggestions for future projects set on human samples.

5. Could you please define better the limitations of your project?

6. Discussion needs to be better organised and re-written on the base of the comments highlighted in this revision.

Thanks for the opportunity to revise this manuscript.

Best regards,

The Reviewer

Reviewer #2 (Remarks to the Author):

Kim et al study the role of TSPYL5 in resistance to radiation and target therapies. The authors nicely demonstrate the mechanism by which TSPYL5 gets activated in response to treatment and its role in activating CSC-associated genes. Although the mechanism is well elucidated, I believe the manuscript lacks demonstration of the role of TSPYL5 in resistance.

MAIN COMMENTS

1- Although the use of TSPYL5-antagonistic peptides is a nice approach, I'm concerned about the specificity of these peptides. Because the peptides competitively bind to AKT, they could also block the effect of AKT on other substrates. In fact, the authors suggest that this could happen in lines 321-322. An inhibition of AKT could also cause the effects observed in vitro and in vivo in figures 8 and 9. These are two key figures of the manuscript. The authors have to demonstrate that the AKT activity is intact, for

example checking the phosphorylation of well known AKT substrates such as PRAS40, GSK3 or FOXO proteins. The proper validation of these peptides is key.

2- In Figure 1 the authors describe the use of H460 resistant cell lines. However, they do not show the level of resistance. In their previous article the authors show that H460 cells have very low levels of phospho-EGFR, which is usually an indication of low sensitivity to EGFR inhibition. How sensitive are the parental cells compared with the resistant?

3- Figure 1 shows the effects of radiation in the expression of CSC-associated genes. However, the authors do not show if the levels of radiation used cause cell death. If they cause cell death, they should explain why there are more cells migrating. From the methods it seems that cells are plated 24 hours after radiation. At this time there will still be cells dying. Similarly, the timing of the western blot in Fig1d is not clear. How long after radiation the cells have been harvested?

4- The authors suggest that the high expression of CSC-associated genes is what is driving resistant to therapy, but they do not completely demonstrate that. To do that, they could treat the ALlow and ALhigh cells isolated in figure 2 and check if they have different sensitivities to radiation or target therapy treatment.

5- Across the manuscript (eg. lines 162 and 178), the authors suggest that first there is an increase of AKT phosphorylation, then increase of TSPYL5 levels and finally reduction of PTEN. However, this kinetics has not been demonstrated.

6- I am quite surprised about the fast reduction of TSPYL5 protein levels after AKT inhibition (Fig4f). Considering the mechanism of protein stability that the authors demonstrate, it is quite surprising that the levels of TSPYL5 are gone just after 10 minutes of treatment with an AKT inhibitor. The authors should discuss that. Moreover, in that same time course it seems that the levels of PTEN protein increase before the levels of RNA. That should also be discussed.

MINOR COMMENTS

1- In Figure 3, does the knock-down of TSPYL5 affects cell viability when cells are grown in 2D with complete media?

2- In Fig 5e the quality of the protein fractionation show be demonstrated checking tubulin in nuclear fraction and HDAC in the cytoplasm.

3- The authors use semi-quantitative RT-PCR to check the levels of expression of the different genes. In the methods they indicate that all the PCRs are done using 30 cycles. Semi-quantitative RT-PCR requires optimization of the cycles used for each gene. Highly expressed genes may need lower cycles to avoid

saturated conditions. How have the authors optimised the RT-PCR protocols?

4- The references of the antibodies used should be added to the manuscript.

5- Supplementary Fig 11a-c are not described in the text.

6- Data from Supplementary Fig 12 should be added in the results section and not in the discussion.

Reviewer #3 (Remarks to the Author):

Summary:

This is a nicely written paper that takes a mechanistic view into the role of TSPYL5 in the formation of CSCs and treatment resistant cell populations in NSCLC. Experimental rationale was clear and conclusions were well supported by the data.

Major Points:

None noted

Minor Points:

Figure 1a – shows a general increase in ALDH1 and CD44 in presumably all cells. Does this argue against the CSC model? Compare to 1e, which does look like a subpopulation of cells responding according to the CSC model.

Figure 1,2,3,4 – images of cells (e.g. 1b,c,f,g) are quite small and hard to evaluate

“Interestingly, in all of these radiation-exposed CSC-like cells with low or high dose, TSPYL5 expression increased in proportion to the radiation dose.” (Lines 107-108) Was TSPYL5 expression increased, or did treatment select for TSPYL5-expressing cells?

Comments for the manuscript COMMSBIO-20-2273

Reviewers' comments:

Reviewer #1 (Remarks to the Author):

This manuscript entitled "Targeting therapy-resistant lung cancer stem cells via disruption of the AKT/TSPYL5/PTEN positive-feedback loop" described the possible identification of testis-specific Y-like protein 5(TSPYL5) as a CSC-associated factor that promotes stemness and epithelial-to-mesenchymal transition in therapy-resistant non-small cell lung cancer (NSCLC) cells.

Beside the limitation regarding the fact that this is developed in cancer cells lines and not in human samples, this manuscript is well-written and methodically well-organized.

• I have some comments for the Authors that need to be addressed:

1. You used ALDH1A1 and ALDH1A3 which are the most common isoforms for targeting cancer stem cells. Then you also used the biomarker CD44 in your manuscript as one of the most used targets for cancer stem cells. I believe that this point needs to be better described and defined, because CD44 is made by several isoforms as well, that you did not described into your manuscript. I believe that this point needs to be better described and defined

Suggested references:

#1> Ranjeeta Thapa and George D. Wilson. The Importance of CD44 as a Stem Cell Biomarker and Therapeutic Target in Cancer. Stem Cells International Volume 2016, Article ID 2087204, 15 pages <http://dx.doi.org/10.1155/2016/2087204>

#2> Hu B, Ma Y, Yang Y, Zhang L, Han H and Chen J: CD44promotes cell proliferation in non-small cell lung cancer. Oncol Lett 15: 5627-5633, 2018

Response:

Thank you very much for your comments on our manuscript. We already knew that several CD44 isoforms exist and the alternative spliced variants play roles in cancer development and progression. Therefore, before proceeding with the experiments, it was examined which type of CD44 isoforms (CD44s or CD44v) is expressed in NSCLC cell lines (A549 and H460) used in this study. It has been confirmed that Pan-CD44 (all CD44 isoforms) is generally used as a marker of EMT or cancer stem cells in lung cancer [1] and CD44s is expressed in A549 cells [2]. In this study, we had used anti-pan-CD44 antibodies, which detect all isoforms of CD44 (see figures below). APC-conjugated CD44 (IM7) Rat mAb (eBioscience™, #17-0441-82) used for the FACS

analysis of CSCs, detects an epitope in the extracellular domain, and is expected to detect all isoforms of CD44. Another CD44-specific antibody [CST#5640 CD44 (8E2) mouse IgG1] used in the immunoblot analysis also detects all isoforms of CD44, of which the epitope is amino acid residues around the intracytoplasmic domain common to all CD44 isoforms. Western blot analysis of A549 or H460 cells revealed a band above 75 kDa, corresponding to the standard form of human CD44 (CD44s). Collectively, CD44 revealed as a CSC biomarker in our study corresponds to CD44s.

As you pointed out, we had not described in detail the isoform of CD44 in the original manuscript. In the revised manuscript, we have described the information of CD44 isoforms (page 18, lane 23-24; page 32, lane 13-14; page 33, lane 17-18).

Ref. 1> Yan Y. et al., Concise Review: Emerging Role of CD44 in Cancer Stem Cells: A Promising Biomarker and Therapeutic Target. *Stem Cells Transl Med.* 2015. 4(9): 1033-1043.

Ref 2> Nishino M. et al., Variant CD44 expression is enriching for a cell population with cancer stem cell-like characteristics in human lung adenocarcinoma. *J Cancer.* 2017. 8(10):1774-1785.

CD44 (8E2) Mouse mAb #5640

<https://www.cellsignal.com/products/primary-antibodies/cd44-8e2-mouse-mab/5640>

<https://www.cusabio.com/Monoclonal-Antibody/CD44-Monoclonal-Antibody-12925746.html>

2. There are a recent manuscript which I would suggest to use, published in 2020 by Masciale V. et al in *Oncotarget* which demonstrated in human NSCLC samples that CD44 and ALDH do not identify the same population of cancer stem cells. In fact, they found that ALDH high cells do not correlate with CD44+ cells but with CD44+/EpCAM+ cells. That is a very important concept, because you could also consider this aspect into your manuscript. CD44+/EPCAM+ cells detect a subpopulation of ALDH high cells in human non-small cell lung cancer: A chance for targeting cancer stem cells?

Suggested reference:

#1> Valentina Masciale, Giulia Grisendi, Federico Banchelli, Roberto D'Amico, Antonino Maiorana, Pamela Sighinolfi, Alessandro Stefani, Uliano Morandi, Massimo Dominici, Beatrice Aramini *Oncotarget.* 2020 Apr 28; 11(17): 1545–1555. Published online 2020 Apr 28. doi: 10.18632/oncotarget.27568

Response:

Thank you for this suggestion. As you noted, Masciale V et al. showed that the CSC-like ALDHhigh cells are highly correlated to CD44+/EPCAM+ cells in human non-small cell lung cancer, suggesting that the use of these markers for future target treatments against lung cancer stem cells. They analyzed the putative CSCs separated from the bulk parental tumor cells, which are a mixture of heterogeneous tumor cells with genetic and epigenetic diversity. In our study, the CSC-like properties of the NSCLC were studied using A549 or H460 NSCLC cell lines, thus limiting the explainable genetic properties of these tumor cells. In fact, A549 cells express EpCAM at a low level (see figure below), which seems not to be a dominant CSC marker in these cells [1, 2].

mRNA expression levels of EpCAM and β -actin as a control were assessed in the indicated carcinoma cell lines by standard RT-PCR (Anticancer Res. 2009.29(5): 1817-1822).

In our studies, we have suggested TSPYL5 as a novel CSC biomarker that regulates the expression of the classical CSC markers, CD44 and ALDH1A1. Further studies on the function of TSPYL5 as a CSC driver are necessary, which may include its effects on the other CSC-related gene expression, including EpCAM.

Ref. 1> Zakaria N et al. Human non-small cell lung cancer expresses putative cancer stem cell markers and exhibits the transcriptomic profile of multipotent cells. BMC Cancer. 2015. 15:84.
Ref. 2> Kim Y. et al. Clinicopathological implications of EpCAM expression in adenocarcinoma of the lung. Anticancer Res. 2009. 29(5): 1817-1822.

3. Did you also consider the isoforms in CD44? If yes, which isoforms? The isoforms consideration is very important for the fact that, despite of ALDH marker, "CD44 is also expressed on healthy cells, making it difficult to be used to specifically differentiate CSCs" (Alhabbab RY).

Please consider these **references**:

Ref#1> Alhabbab RY. Targeting Cancer Stem Cells by Genetically Engineered Chimeric Antigen Receptor T Cells. Front Genet. 2020;11:312. Published 2020 Apr 22. doi:10.3389/fgene.2020.00312

Ref#2> Chen C, Zhao S, Karnad A, Freeman JW. The biology and role of CD44 in cancer progression: therapeutic implications. J Hematol Oncol. 2018; 11(1):64. Published 2018 May 10. doi:10.1186/s13045-018-0605-5

Response:

I agree with your suggestion. Several experimentally verified CD44v forms have been shown to be directly involved in many malignant tumors and some correlate with metastatic progression [1]. In non-small cell lung carcinoma, the upregulation of CD44v6 correlates with metastasis and poor prognosis [2-4]. However, there have been no reports on the expression of the CD44 variant in A549 or H460 cell lines, which has been shown to express CD44s [5-7].

To confirm the CD44 isoforms in these cells, we have sequenced the RT-PCR product amplified from cDNA of A549 or H460 cells. RT-PCR primers for CD44, which correspond to exon 4 and exon 15 (forward: 5'-CCAATGCCTTTGATGGACCA-3', reverse: 5'-TGTGAGTGTCCATCTGATTC-3'), can amplify six isoforms of CD44 with different product size.

Target	Product length (bp)	Products on target templates			
NM_000610.4 Homo sapiens CD44 molecule (Indian blood group) (CD44), transcript variant 1, mRNA	1477	F_primer	1	CCAATGCCTTTGATGGACCA	20
		Template	540	559
		R_primer	1	TGTGAGTGTCCATCTGATTC	20
		Template	2016	1997
NM_001001389.2 Homo sapiens CD44 molecule (Indian blood group) (CD44), transcript variant 2, mRNA	1348	F_primer	1	CCAATGCCTTTGATGGACCA	20
		Template	540	559
		R_primer	1	TGTGAGTGTCCATCTGATTC	20
		Template	1887	1868
NM_001001390.2 Homo sapiens CD44 molecule (Indian blood group) (CD44), transcript variant 3, mRNA	730	F_primer	1	CCAATGCCTTTGATGGACCA	20
		Template	540	559
		R_primer	1	TGTGAGTGTCCATCTGATTC	20
		Template	1269	1250
NM_001001391.2 Homo sapiens CD44 molecule (Indian blood group) (CD44), transcript variant 4, mRNA	334	F_primer	1	CCAATGCCTTTGATGGACCA	20
		Template	540	559
		R_primer	1	TGTGAGTGTCCATCTGATTC	20
		Template	873	854
NM_001202555.2 Homo sapiens CD44 molecule (Indian blood group) (CD44), transcript variant 6, mRNA	538	F_primer	1	CCAATGCCTTTGATGGACCA	20
		Template	540	559
		R_primer	1	TGTGAGTGTCCATCTGATTC	20
		Template	1077	1058
NM_001202557.2 Homo sapiens CD44 molecule (Indian blood group) (CD44), transcript variant 8, mRNA	334	F_primer	1	CCAATGCCTTTGATGGACCA	20
		Template	540	559
		R_primer	1	TGTGAGTGTCCATCTGATTC	20
		Template	873	854

RT-PCR products analyzed by agarose gel have shown DNA product with a size of 334 bp as a major product and smear bands above the major band (see figure below). We had extracted these RT-PCR products and cloned them into T-vector to confirm their sequences. As expected, the major product of 334bp is corresponding to transcript variant 4 or variant 8. Also, PCR product of 537bp has been confirmed as CD44 transcript variant 6.

CD44 primers	Forward	5'-CCAATGCCTTTGATGGACCA-3'
	Reverse	5'-TGTGAGTGCCATCTGATTC-3'

Sequencing (#7, 12, 20, 21)

Clone No.	RT-PCR product		Matched sequences	
	sequences	size	Reference transcript	Localization (size)
#7	CAATGCCCTTTGATGGACCAATTACCATAACTATTGTTAAACGGTATGGCACCCGCTATGTC CAGAAAGGAGAATACAGAACGAATCCTGAAGACATCTACCCAGCAACCTACTGATGATG ACCTGAGCAGCGGCTCCTCCAGTGAAGGAGCAGCACTTCAGGAGGTTACATCTTTTACAC CTTTTCTACTGTACACCCATCCAGACGAAGACAGTCCCTGGATCACCAGCACAGAC AGAATCCCTGCTACCAATAGGAATGATGTCACAGGTGAAGAGAGACC CAAATCATTCTG AAGGCTCAACTACTTTACTGGAAGTTATACCTCTCATTACCCACAGCAAGGAAAGCAG GACCTTCATCCAGTACCTCAGCTAAGACTGGGCTTTGGAGTTACTGACGTTACTGTT CGAGATTCCAACCTAATGTC AATCGTTCTTATCAGGAGACCAAGACACATTCCACCCCA GTGGGGGTCATACCCACTCATGGATCTGAAATCAGATGGACACTCACA	537bp	NM_001202555.2. Homo sapiens CD44 molecule, transcript variant 6, mRNA,	541..1077 (537bp)
#12	CCAATGCCCTTTGATGGACCAATTACCATAACTATTGTTAAACGGTATGGCACCCGCTATGTC CAGAAAGGAGAATACAGAACGAATCCTGAAGACATCTACCCAGCAACCTACTGATGATG ACCTGAGCAGCGGCTCCTCCAGTGAAGGAGCAGCACTTCAGGAGGTTACATCTTTTACAC CTTTTCTACTGTACACCCATCCAGACGAAGACAGTCCCTGGATCACCAGCACAGAC AGAATCCCTGCTACCCAGACCAAGACACATTCCACCCAGTGGGGGTCATACCCACT CATGGATCTGAAATCAGATGGACACTCACA	334 bp	NM_001001391.2. Homo sapiens CD44 molecule, transcript variant 4, mRNA NM_001202557.2. Homo sapiens CD44 molecule, transcript variant 8, mRNA	540..873 (334 bp) 540..873 (334 bp)
#20	CCAATGCCCTTTGATGGACCAATTACCATAACTATTGTTAAACGGTATGGCACCCGCTATGTC CAGAAAGGAGAATACAGAACGAATCCTGAAGACATCTACCCAGCAACCTACTGATGATG ACCTGAGCAGCGGCTCCTCCAGTGAAGGAGCAGCACTTCAGGAGGTTACATCTTTTACAC CTTTTCTACTGTACACCCATCCATACCCACTCATGGATCTGAAATCAGATGGACACTCACA	245 bp	No transcripts fully matched	
#21	CCAATGCCCTTTGATGGACCAATTACCATAACTATTGTTAAACGGTATGGCACCCGCTATGTC CAGAAAGGAGAATACAGAACGAATCCTGAAGACATCTACCCAGCAACCTACTGATGATG ACCTGAGCAGCGGCTCCTCCAGTGAAGGAGCAGCACTTCAGGAGGTTACATCTTTTACAC CTTTTCTACTGTACACCCATCCAGACGAAGACAGTCCCTGGATCACCAGCACAGAC AGAATCCCTGCTACCCAGACCAAGACACATTCCACCCAGTGGGGGTCATACCCACT CATGGATCTGAAATCAGATGGACACTCACA	334 bp	NM_001001391.2. Homo sapiens CD44 molecule, transcript variant 4, mRNA NM_001202557.2. Homo sapiens CD44 molecule, transcript variant 8, mRNA	540..873 (334 bp) 540..873 (334 bp)

* RT-PCR primer sequences were indicated in red characters.

The CD44 transcript variant 4 (NM_001001391.2) is composed of ten CD44 exons (exon 1 – exon 5, exon 15 – exon 19), which express CD44 standard (CD44s). The CD44 transcript variant 8 is also amplified with the same primers; however, its product has a short cytoplasmic tail, which cannot be detected by western blot analysis in our study. Collectively, it has been confirmed again that A549 or H460 cells express CD44s as a major isoform of CD44.

The CD44 transcript variant 6 has also been detected in A549 cells, although its expression is very low (see figure above; unpublished result). It contains ten exons of CD44 transcript variant 4 as well as exon 14 additionally, which corresponds to CD44v10 (see figure below).

(A) CD44 gene consists of 20 exons

(B) CD44s lacks the entire variable region

(C) CD44v6 contains variant exon 6

http://atlasgeneticsoncology.org/Genes/GC_CD44.html

CD44v10 has been known to interact with Rho-kinase (ROK), which activates inositol 1,4,5-triphosphate (IP3) receptor-mediated Ca²⁺ signaling during hyaluronan (HA)-induced endothelial cell migration [8]. CD44v10 expression is also related to tumor aggressiveness in oral cancer or melanoma [9, 10]. On these backgrounds, it is expected that CD44v10 contributes to the tumorigenicity of other types of tumor. Also, the anti-CD44v10 antibody has been used for therapeutic purposes on leukemia [11]. However, so far, there have been no studies on CD44v10 expression in NSCLC. Further researches on CD44 variants, including CD44v10, expressed in NSCLC is necessary, especially on its contribution to CSC properties.

There have been errors in the **RT-PCR primer sequences of CD44 in the revised Supplementary Table 6** of the original manuscript, so we corrected it as follows.

Target gene	Primer sequences (5' → 3') _in original manuscript	Primer sequences (5' → 3') _in revised manuscript
CD44	forward: TTCAACCTCGAATAAAAAGCTGC	forward: CCAATGCCTTTGATGGACCA
	reverse: TATTCAAATCGATCTGCGCC	reverse: TGTGAGTGTCCATCTGATTC

Ref. 1> Thapa R et al., The Importance of CD44 as a Stem Cell Biomarker and Therapeutic Target in Cancer. *Stem Cells Int.* 2016; 2016: 2087204.

Ref. 2> Afify A.M. et al., Expression of CD44s and CD44v6 in lung cancer and their correlation with prognostic factors. *Int J Biol Markers.* 2011. 26(1): 50-57.

Ref. 3> Nguyen V.N. et al., CD44 and its v6 spliced variant in lung carcinomas: relation to NCAM, CEA, EMA and UPI and prognostic significance. *Neoplasma.* 2000. 47(6): 400-408.

Ref. 4> Hirata T. et al., Expression of CD44 variant exon 6 in stage I non-small cell lung carcinoma as a prognostic factor. *Cancer Res.* 1998. 58 (6):1108-1110.

Ref. 5> Horibe S. et al., CD44v-dependent upregulation of xCT is involved in the acquisition of cisplatin-resistance in human lung cancer A549 cells. *Biochem Biophys Res Commun.* 2018. 507 (1-4): 426-432.

Ref. 6> Hu B., CD44 promotes cell proliferation in non-small cell lung cancer. *Oncol Lett.* 2018. 15 (4): 5627-5633.

Ref. 7> Nishino M., Variant CD44 expression is enriching for a cell population with cancer stem cell-like characteristics in human lung adenocarcinoma. *J Cancer.* 2017. 8 (10): 1774-1785.

Ref. 8> Singleton P.A. et al., CD44v10 interaction with Rho-kinase (ROK) activates inositol 1,4,5-triphosphate (IP3) receptor-mediated Ca²⁺ signaling during hyaluronan (HA)-induced endothelial cell migration. *Cell Motil Cytoskeleton*. 2002. 53(4): 293-316.
Ref. 9> Shah K. et al., Uncovering the potential of CD44v/SYNE1/miR34a axis in salivary fluids of oral cancer patients. *J Oral Pathol Med*. 2018. 47(4): 345-352.
Ref. 10> Yoshinari C. et al., CD44 variant isoform CD44v10 expression of human melanoma cell lines is upregulated by hyaluronate and correlates with migration. *Comparative Study Melanoma Res*. 1999. 9(3): 223-231.
Ref. 11> Erb U. et al., CD44 standard and CD44v10 isoform expression on leukemia cells distinctly influences niche embedding of hematopoietic stem cells. *J Hematol Oncol*. 2014. 7:29.

4. *With regards of the methods, it needs that you better specify which are the histopathological subtypes of NSCLC cells lines you used into your experiments. That is better for the reader, as well as also for giving the suggestions for future projects set on human samples.*

Response:

Thank you for pointing this out. NSCLC is classified into three histopathological subtypes: adenocarcinoma (AD), squamous cell carcinoma (SQ), and large cell carcinoma (LC). A549 (ATCC® CCL-185™) is a human lung adenocarcinoma cell line; NCI-H460 [H460] (ATCC® HTB-177™) is a human large cell lung cancer. The descriptions of the cell lines were added to the revised manuscript as suggested by the reviewer (page 4, lane 26).

5. *Could you please define better the limitations of your project?*

Response:

In our study, human lung cancer cell lines were used as in vitro models for the analysis of CSC properties. Chemoradiotherapy-resistant A549 cells were used as CSC-like cells, and therapy-sensitive H460 cells were used as comparative tumor cells, of which the results have suggested that TSPYL5 is a novel upstream regulator of CSCs. For the development of CSC-targeting molecule using TSPYL5 peptides, studies on the clinical samples (eg. NSCLC tissue/cells) to confirm the expression of TSPYL5 and its correlation to CSC properties are necessary.

6. *Discussion needs to be better organised and re-written on the base of the comments highlighted in this revision.*

Response:

We have revised the discussion to reflect your suggestions. Thanks for the opportunity to revise this manuscript.

Reviewer #2 (Remarks to the Author):

Kim et al study the role of TSPYL5 in resistance to radiation and target therapies. The authors nicely

demonstrate the mechanism by which TSPYL5 gets activated in response to treatment and its role in activating CSC-associated genes. Although the mechanism is well elucidated, I believe the manuscript lacks demonstration of the role of TSPYL5 in resistance.

Response:

I agree with this. CSC resistance is defined by intrinsic as well as extrinsic factors [1]. As intrinsic factors, EMT, oxidative modulators, metabolic plasticity, stemness capacity, signaling (PI3K/AKT, NF- κ B, etc.) were suggested. In our study, TSPYL5 has been suggested as a transcriptional regulator of CD44, ALDH1A1, and PTEN, which are intrinsic CSC resistance drivers. Collectively, TSPYL5 acts as an upstream regulator of CSC resistance drivers. The description on the role of TSPYL5 in resistance is added to the revised Abstract.

Ref 1> Najafi M. et al., Cancer stem cell (CSC) resistance drivers. Life Sci. 2019 Oct 1; 234:116781.

MAIN COMMENTS

1- *Although the use of TSPYL5-antagonistic peptides is a nice approach, I'm concerned about the specificity of these peptides. Because the peptides competitively bind to AKT, they could also block the effect of AKT on other substrates. In fact, the authors suggest that this could happen in lines 321-322. An inhibition of AKT could also cause the effects observed in vitro and in vivo in figures 8 and 9. These are two key figures of the manuscript. The authors have to demonstrate that the AKT activity is intact, for example checking the phosphorylation of well-known AKT substrates such as PRAS40, GSK3 or FOXO proteins. The proper validation of these peptides is key.*

Response:

Thank you for pointing this out. TSPYL5-antagonistic peptide sequence includes AKT substrate-like sequence motif (RXRXXS*/T), which may bind to substrate binding site. Therefore, TSPYL5-antagonistic peptide would competitively inhibit various AKT substrate proteins, which can be demonstrated by in vitro AKT activity assay using TSPYL5-antagonistic peptide as an inhibitor. When the cells are treated with TSPYL5-antagonistic peptide, it would competitively inhibit AKT phosphorylation of TSPYL5 as well as other AKT substrates depending on the affinity to AKT. However, the inhibitory effect of TS peptide in cells is not so simple. In the TSPYL5-expressing NSCLC cells, TS-peptide would trigger the degradation of TSPYL5, which induces PTEN expression and consequently suppress AKT activity in a positive feedback manner. Consequently, downregulated AKT activity may show the effects on other AKT substrate proteins, such as GSK3 β (as shown in the figure below).

Western blot analysis of A549 cells treated with TS-peptides (10 μ M, 12 hrs).

Contrary to the effects on A549 cells, TSPYL5-deficient H460 cells did not show any cellular response to TS-peptides, which implicates that the inhibitory effects on the other AKT substrates are not so effective in these cells (Supplementary Fig. 9). These results implicate the effect of TS-antagonistic peptide on AKT inhibition in A549 cells seems to be not so effective as a simple competitive inhibitor but driven by a positive feedback AKT/PTEN loop resulting in an accumulative affect only in TSPYL5-expressing cells. AKT inhibition mechanism by TS-peptides also differs from that of the conventional AKT inhibitors, ATP-competitive or allosteric. In further studies, we would study the characteristics of TS-peptides binding to AKT and effects on the other AKT substrates in detail.

2- In Figure 1 the authors describe the use of H460 resistant cell lines. However, they do not show the level of resistance. In their previous article the authors show that H460 cells have very low levels of phospho-EGFR, which is usually an indication of low sensitivity to EGFR inhibition. How sensitive are the parental cells compared with the resistant?

Response:

Thank you for pointing this out. There have been several studies on the effects of gefitinib in H460 cells [1-3]. These studies showed the IC50 value of H460 cells to gefitinib as ~8 μ M [1]. According to these results, gefitinib-resistant (GR) H460 cells were generated by the stepwise escalation of gefitinib concentration from 1 to 10 μ M over 4 months. The references on the gefitinib effect on H460 cells were added in the revised manuscript (page 5, lane 14).

 Ref. 1> Choi K et al., Transcriptional Profiling of Non-Small Cell Lung Cancer Cells with Activating EGFR Somatic Mutations. PLoS ONE. 2007. 2(11): e1226.
 Ref. 2> Yang T., et al. LncRNA MALAT1 Depressed Chemo-Sensitivity of NSCLC Cells through Directly Functioning on miR-197-3p/p120 Catenin Axis. Mol Cells. 2019. 42(3): 270-283.
 Ref. 3> Hsia T-C. et al., Phenethyl Isothiocyanate Induces Apoptotic Cell Death Through the Mitochondria-dependent Pathway in Gefitinib-resistant NCI-H460 Human Lung Cancer Cells In Vitro. Anticancer Res. 2018.38(4): 2137-2147.

3- Figure 1 shows the effects of radiation in the expression of CSC-associated genes. However, the authors do not show if the levels of radiation used cause cell death. If they cause cell death, they should explain why there are more cells migrating. From the methods it seems that cells are plated 24 hours after radiation. At this time there will still be cells dying. Similarly, the timing of the western blot in Fig1d is not clear. How long after radiation the cells have been harvested?

Response:

Thank you for pointing this out. Irradiation with a single low dose of 2 Gy or 4 Gy to A549 cells did not cause severe cell death. Most cells (> 90%) survive after a low dose of irradiation, while it may cause some inhibition of proliferation. These effects were already reported in our previous paper [1]. In the experiment in Fig 1d, cells were collected 24 hours after irradiation and analyzed for the western blot analysis.

To induce cell death in A549 cells, high-dose radiation (> 6 Gy) must be irradiated, as shown in Fig. 9a and b. Radiation-induced cell death of A549 cells had already been reported in our previous study [2]. Cell viability in these cells decreased to less than 60 %.

Ref 1> Kim IG et al., Disturbance of DKK1 level is partly involved in survival of lung cancer cells via regulation of ROMO1 and gamma-radiation sensitivity. *Biochem Biophys Res Commun.* 2014. 443(1): 49-55.

Ref 2> Lee JH et al., Tescalcin/c-Src/IGF1R β -mediated STAT3 activation enhances cancer stemness and radioresistant properties through ALDH1. *Sci Rep.* 2018. 8(1): 10711.

4- The authors suggest that the high expression of CSC-associated genes is what is driving resistant to therapy, but they do not completely demonstrate that. To do that, they could treat the AL low and AL high cells isolated in figure 2 and check if they have different sensitivities to radiation or target therapy treatment.

Response:

Thank you for pointing this out. The radio-resistance of ALDH+ CSC-like cells sorted from A549 cells had been analyzed in our previous study [1]. Besides our study, a number of researches have shown that ALDH^{high} cells have enhanced resistance to radiation [2-7].

Ref. 1> Kim IG et al., Fibulin-3-mediated inhibition of epithelial-to-mesenchymal transition and self-renewal of ALDH+ lung cancer stem cells through IGF1R signaling. *Oncogene.* 2014. 33(30): 3908-3917.

Ref. 2> Liu L et al. ALDH1A1 Contributes to PARP Inhibitor Resistance via Enhancing DNA Repair in BRCA2 -/- Ovarian Cancer Cells. *Mol Cancer Ther.* 2020. 19(1): 199-210.

Ref. 3> Dehghan Harati M et al. Nanog Signaling Mediates Radioresistance in ALDH-Positive Breast Cancer Cells. *Int J Mol Sci.* 2019. 20(5): 1151.

Ref. 4> Han SY et al. Marsdenia tenacissima extract restored gefitinib sensitivity in resistant non-small cell lung cancer cells. *Lung Cancer.* 2012. 75(1): 30-37.

Ref. 5> Martín M et al. Aldehyde dehydrogenase isoform 1 (ALDH1) expression as a predictor of radiosensitivity in laryngeal cancer. *Clin Transl Oncol.* 2016. 18(8): 825-830.

Ref. 6> Cojoc M et al. Aldehyde Dehydrogenase Is Regulated by β -Catenin/TCF and Promotes Radioresistance in Prostate Cancer Progenitor Cells. *Cancer Res.* 2015. 75(7):1482-1494.

Ref. 7> Qu Y et al. Antitumor activity of selective MEK1/2 inhibitor AZD6244 in combination with PI3K/mTOR inhibitor BEZ235 in gefitinib-resistant NSCLC xenograft models. *J Exp Clin Cancer Res.* 2014. 33(1): 52.

5- Across the manuscript (eg. lines 162 and 178), the authors suggest that first there is an increase of AKT phosphorylation, then increase of TSPYL5 levels and finally reduction of PTEN. However, this kinetics has not been demonstrated.

Response:

Although we had not performed a detailed time-dependent analysis on TSPYL5-mediated regulation of CSCs, we had suggested the regulatory mechanism based on experiments with AKT inhibitors (revised Fig. 4f: see Figure below). AKT inactivation by MK2206 induced degradation of TSPYL5, whereas PTEN expression was increased at the transcript as well as protein levels.

In this paper, we have focused on presenting new CSC-related functions of TSPYL5. Further study is necessary on the overall mechanism of TSPYL5-dependent cancer stemness, including how TSPYL5 gene expression is regulated.

6- I am quite surprised about the fast reduction of TSPYL5 protein levels after AKT inhibition (Fig 4f). Considering the mechanism of protein stability that the authors demonstrate, it is quite surprising that the levels of TSPYL5 are gone just after 10 minutes of treatment with an AKT inhibitor. The authors should discuss that. Moreover, in that same time course it seems that the levels of PTEN protein increase before the levels of RNA. That should also be discussed.

Response:

Thank you for pointing this out. The rapid degradation of TSPYL5 induced by AKT inhibitors is the most important result of this study. This evidence came from repeating the experiments independently several times, with similar results. However, the results showing the degradation of

TSYPYL5 in the original manuscript are the WB results exposed for a short time, making the reviewer confused. We repeated the WB analysis (see Fig A) and replaced the revised manuscript's result as follows (revised Fig 4f: see Fig B).

Also, the result of the PTEN transcript has been revised, as the reviewer pointed out. In the original manuscript, the transcript of PTEN of NT (not-treated), a control sample, was not shown because of the image's inappropriate brightness. The image's brightness has been adjusted to appear the PTEN product in the NT lane, and the re-adjusted image replaced the original image.

MINOR COMMENTS

1- In Figure 3, does the knock-down of TSPYL5 affects cell viability when cells are grown in 2D with complete media?

Response:

In our previous study on TSPYL5 [1], the colony-forming assay for the TSPYL5 knock-down A549 cells was performed, showing cell growth inhibition (see Figure below). The colony-forming assay was performed with complete media in 2D.

 Ref. 1> Eun Jin Kim et al., TSPYL5 is involved in cell growth and the resistance to radiation in A549 cells via the regulation of p21(WAF1/Cip1) and PTEN/AKT pathway. Biochem. Biophys. Res Commun. 2010. 392(3):448-453.

Eun Jin Kim et al., TSPYL5 is involved in cell growth and the resistance to radiation in A549 cells via the regulation of p21(WAF1/Cip1) and PTEN/AKT pathway. *Biochem. Biophys. Res Commun.* 2010. 392(3):448-453.

2- In Fig 5e the quality of the protein fractionation show be demonstrated checking tubulin in nuclear fraction and HDAC in the cytoplasm.

Response:

As you pointed out, it is necessary to determine the fractionation quality by identifying tubulin and HDAC in the nuclear fraction and cytoplasm fraction. The cytoplasmic and nuclei fractions were examined by WB analysis (see figure below). Each cell's subcellular fractions showed cross-contamination; however, each cell's fractionation yield is similar, which may offset the contamination effects. In Fig 5e, HDAC and tubulin were used to control the loading amount of each fraction.

3- The authors use semi-quantitative RT-PCR to check the levels of expression of the different genes. In the methods they indicate that all the PCRs are done using 30 cycles. Semi-quantitative RT-PCR requires optimization of the cycles sed for each gene. Highly expressed genes may need lower cycles to avoid saturated conditions. How have the authors optimized the RT-PCR protocols?

Response:

Thank you for pointing this out. The amplification cycles were optimized for each PCR reaction,

which was adjusted between 25-30 cycles, and comparable results were presented as the final results. The RT-PCR method has been revised (page 20, lane 1).

4- *The references of the antibodies used should be added to the manuscript.*

Response:

Thank you for pointing this out. As you suggested, the information on Antibodies has been added to the revised manuscript in the Supplementary Table 4.

5- *Supplementary Fig 11a-c are not described in the text.*

Response:

Supplementary Fig 11 has been revised and the descriptions on the results were added to the results (page 14, lane 23, 26).

6- *Data from Supplementary Fig 12 should be added in the results section and not in the discussion.*

Response:

As you suggested, the description of Supplementary Fig 12 has been added at the end of the result section (page 15, lane 4-7).

Reviewer #3 (Remarks to the Author):

Summary:

This is a nicely written paper that takes a mechanistic view into the role of TSPYL5 in the formation of CSCs and treatment resistant cell populations in NSCLC. Experimental rationale was clear and conclusions were well supported by the data.

Major Points:

None noted

Minor Points:

Figure 1a – shows a general increase in ALDH1 and CD44 in presumably all cells. Does this argue against the CSC model? Compare to 1e, which does look like a subpopulation of cells responding according to the CSC model.

Response:

Thank you for pointing this out. The cancer stem cell model, also known as the Hierarchical Model, proposes that tumors are hierarchically organized [1]. This model suggests that only certain subpopulations of cancer stem cells have the ability to drive the progression of cancer, meaning that there are specific (intrinsic) characteristics that can be identified. According to the CSC model, CSC

identity is hardwired, and there is limited plasticity in the tumor hierarchy. Although the evidence has been accumulated that supports the existence of CSC hierarchies in many prevalent tumor types, there is also an increasing appreciation that not every cancer adheres to such a CSC model. Several studies have provided evidence that both CSCs and non-CSCs are plastic and capable of undergoing phenotypic transitions in response to appropriate stimuli [2].

As pointed out by the reviewer, the CSC characteristics of A549 cells differ from that of H460 cells. In A549 cells, CSC biomarker expression had increased by the repeated low dose radiation without cell death. On the other hand, long-term treatment with high dose gefitinib induced cell death in H460 cells and selected the survived cells, which showed enhanced CSC characteristics. These results implicate that the selection of CSC cells from H460 cells by drug treatment is well-matched with the CSC hierarchical model. On the other hand, A549 cells seem to show the plasticity for the phenotype transition to CSC induced by external stimulation.

These results are explainable based on the expression of TSPYL5, a novel CSC regulator proposed in our manuscript. In a previous study [3], we had confirmed that TSPYL5 expression is regulated by promoter methylation. A549 cells maintain TSPYL5 promoter methylation at a low level, which possesses a high potential of TSPYL5 expression. When the AKT activity increases in A549 cells due to external stimuli such as radiation, AKT-dependent phosphorylation increases TSPYL5 stabilization and consequently increases CSC properties. In contrast, H460 cells maintain a high level of TSPYL5 promoter methylation, and they seem to rapidly die upon external stimuli such as radiation because of disabling to increase TSPYL5 expression. However, the cells that survived even after radiation or gefitinib treatment showed increased TSPYL5, which is presumed to be selected cells with a low level of TSPYL5 promoter methylation. This can be confirmed in future studies.

Ref 1> Bonnet D, Dick JE. Human acute myeloid leukemia is organized as a hierarchy that originates from a primitive hematopoietic cell. Nat Med. 1997. 3(7):730-737.

Ref 2> Qin S et al, Emerging role of tumor cell plasticity in modifying therapeutic response. Signal Transduct Target Ther. 2020.5(1):228.

Ref 3> Kim EJ et al. TSPYL5 is involved in cell growth and the resistance to radiation in A549 cells via the regulation of p21(WAF1/Cip1) and PTEN/AKT pathway. Biochem Biophys Res Commun. 2010. 392(3):448-453.

Figure 1,2,3,4 – images of cells (e.g.1b,c,f,g) are quite small and hard to evaluate

Response:

Thank you for this suggestion. As you suggested, the image size was adjusted larger in the revised manuscript.

“Interestingly, in all of these radiation-exposed CSC-like cells with low or high dose, TSPYL5 expression increased in proportion to the radiation dose.” (Lines107-108) Was TSPYL5 expression increased, or did treatment select for TSPYL5-expressing cells?

Response:

As described above, we can explain these cellular events in the A549 cells by suggesting TSPYL5 as a CSC driver. In A549 cells, TSPYL5 maintains a low DNA methylation level and is in the transcriptionally active status. TSPYL5 protein is stabilized by AKT-dependent phosphorylation, and AKT activation by external stimuli, such as a low dose of radiation reinforces the TSPYL5-mediated positive feedback loop that enhances CSC characteristics. Even when most cells die after a high dose of radiation stimuli (20 Gy), A549 cells expressing a high level of TSPYL5 exhibit radiation-resistant CSC characteristics, which survive and finally maintain a new group of tumor cells.

REVIEWERS' COMMENTS:

Reviewer #1 (Remarks to the Author):

Thanks for answering to my questions in a focused and constructive approach.

I believe that your manuscript can be published in this way.

Best regards

Reviewer #2 (Remarks to the Author):

The authors have addressed all the issues that I raised.

Reviewer #3 (Remarks to the Author):

All concerns have been satisfactorily addressed.